# Towards Better Dynamic Graph Learning: New Architecture and Unified Library

**Le Yu, Leilei Sun,**\* **Bowen Du, Weifeng Lv**
State Key Laboratory of Software Development Environment
School of Computer Science and Engineering
Beihang University
{yule,leileisun,dubowen,lwf}@buaa.edu.cn

## Abstract

We propose DyGFormer, a new Transformer-based architecture for dynamic graph learning. DyGFormer is conceptually simple and only needs to learn from nodes' historical first-hop interactions by: (i) a neighbor co-occurrence encoding scheme that explores the correlations of the source node and destination node based on their historical sequences; (ii) a patching technique that divides each sequence into multiple patches and feeds them to Transformer, allowing the model to effectively and efficiently benefit from longer histories. We also introduce DyGLib, a unified library with standard training pipelines, extensible coding interfaces, and comprehensive evaluating protocols to promote reproducible, scalable, and credible dynamic graph learning research. By performing exhaustive experiments on thirteen datasets for dynamic link prediction and dynamic node classification tasks, we find that DyGFormer achieves state-of-the-art performance on most of the datasets, demonstrating its effectiveness in capturing nodes' correlations and long-term temporal dependencies. Moreover, some results of baselines are inconsistent with previous reports, which may be caused by their diverse but less rigorous implementations, showing the importance of DyGLib. All the used resources are publicly available at `https://github.com/yule-BUAA/DyGLib`.

## 1 Introduction

Dynamic graphs denote entities as nodes and represent their interactions as links with timestamps [26], which can model many real-world scenarios such as social networks [28, 50, 2], user-item interaction systems [31, 15, 71, 72, 70], traffic networks [67, 63, 19, 4, 69], and physical systems [24, 47, 43]. In recent years, representation learning on dynamic graphs has become a trending research topic [26, 49, 65]. There are two main categories of existing methods: discrete-time [39, 17, 48, 11, 66] and continuous-time [64, 45, 60, 51, 12]. In this paper, we focus on the latter approaches because they are more flexible and effective than the formers and are being increasingly investigated.

Despite the rapid development of dynamic graph learning methods, they still suffer from two limitations. Firstly, most of them independently compute the temporal representations of nodes in an interaction without exploiting nodes' correlations, which are often indicative of future interactions. Moreover, existing methods learn at the interaction level and thus only work for nodes with fewer interactions. When nodes have longer histories, they require sampling strategies to truncate the interactions for feasible calculations of the computationally expensive modules like graph convolutions [55, 64, 45, 36, 9, 59], temporal random walks [60, 25] and sequential models [57, 12]. Though some approaches use memory networks [61, 53] to sequentially process interactions with affordable computational costs [28, 55, 45, 36, 59, 35], they are faced with the vanishing/exploding gradients

---

\*Corresponding Author.

37th Conference on Neural Information Processing Systems (NeurIPS 2023).

due to the usage of recurrent neural networks [40, 57]. Therefore, we conclude that *previous methods lack the ability to capture either nodes' correlations or long-term temporal dependencies*.

Secondly, the training pipelines of different methods are inconsistent and often lead to poor reproducibility. Also, existing methods are implemented by diverse frameworks (e.g., Pytorch [41], Tensorflow [1], DGL [58], PyG [16], C++), making it time-consuming and difficult for researchers to quickly understand the algorithms and further dive into the core of dynamic graph learning. Although there exist some libraries for dynamic graph learning [18, 46, 74], they mainly focus on dynamic network embedding methods [18], discrete-time graph learning methods [46], or engineering techniques for training on large-scale dynamic graphs [74] (elaborated in Section 2). Currently, we find that *there are still no standard tools for continuous-time dynamic graph learning*.

In this paper, we aim to address the above drawbacks with two key technical contributions.

**We propose a new Transformer-based dynamic graph learning architecture (DyGFormer)**. DyGFormer is conceptually simple by solely learning from the sequences of nodes' historical first-hop interactions. To be specific, DyGFormer is designed with a neighbor co-occurrence encoding scheme, which encodes the appearing frequencies of each neighbor in the sequences of the source and destination nodes to explicitly explore their correlations. In order to capture long-term temporal dependencies, DyGFormer splits each node's sequence into multiple patches and feeds them to Transformer [56]. This patching technique not only makes the model effectively benefit from longer histories via preserving local temporal proximities, but also efficiently reduces the computational complexity to a constant level that is irrelevant to the input sequence length.

**We present a unified continuous-time dynamic graph learning library (DyGLib)**. DyGLib is an open-source toolkit with standard training pipelines, extensible coding interfaces, and comprehensive evaluating strategies, aiming to foster standard, scalable, and reproducible dynamic graph learning research. DyGLib has integrated a variety of continuous-time dynamic graph learning methods as well as benchmark datasets from various domains. It trains all the methods via the same pipeline to eliminate the influence of different implementations and adopts a modularized design for developers to conveniently incorporate new datasets and algorithms based on their specific requirements. Moreover, DyGLib supports both dynamic link prediction and dynamic node classification tasks with exhaustive evaluating strategies to provide comprehensive comparisons of existing methods.

To evaluate the model performance, we conduct extensive experiments based on DyGLib, including dynamic link prediction under transductive and inductive settings with three negative sampling strategies as well as dynamic node classification. From the results, we observe that: (i) DyGFormer outperforms existing methods on most datasets, demonstrating its superiority in capturing nodes' correlations and long-term temporal dependencies; (ii) some findings of baselines are not in line with previous reports because of their varied pipelines and problematic implementations, which illustrates the necessity of introducing DyGLib. We also provide an in-depth analysis of the neighbor co-occurrence encoding and patching technique for a better understanding of DyGFormer.

## 2   Related Work

**Dynamic Graph Learning**. Representation learning on dynamic graphs has been widely studied in recent years [26, 49, 65]. Discrete-time methods manually divide the dynamic graph into a sequence of snapshots and apply static graph learning methods on each snapshot, which ignore the temporal order of nodes in each snapshot [39, 17, 48, 11, 66]. In contrast, continuous-time methods directly learn on the whole dynamic graph with temporal graph neural networks [55, 64, 45, 36, 9, 59], memory networks [28, 55, 45, 36, 59, 35], temporal random walks [60, 25] or sequential models [57, 12]. Although insightful, most existing dynamic graph learning methods neglect the correlations between two nodes in an interaction. They also fail to handle nodes with longer interactions due to unaffordable computational costs of complex modules or issues in optimizing models (e.g., the vanishing/exploding gradients). In this paper, we propose a new Transformer-based architecture to show the necessity of capturing nodes' correlations and long-term temporal dependencies, which is achieved by two designs: a neighbor co-occurrence encoding scheme and a patching technique.

**Transformer-based Applications in Various Fields**. Transformer [56] is an innovative model that employs the self-attention mechanism to handle sequential data, which has been successfully applied in a variety of domains, such as natural language processing [13, 33, 6], computer vision [7, 14, 34]

and time series forecasting [30, 73, 62]. The idea of dividing the original data into patches as inputs of the Transformer has been attempted in some studies. ViT [14] splits an image into multiple patches and feeds the sequence of patches' linear embeddings into a Transformer, which achieves surprisingly good performance on image classification. PatchTST [38] divides a time series into subseries-level patches and calculates the patches by a channel-independent Transformer for long-term multivariate time series forecasting. In this work, we propose a patching technique to learn on dynamic graphs, which can provide our approach with the ability to handle nodes with longer histories.

**Graph Learning Library**. Currently, there exist many libraries for static graphs [5, 58, 16, 22, 8, 29, 32], but few for dynamic graph learning [18, 46, 74]. DynamicGEM [18] focuses on dynamic graph embedding methods, which just consider the graph topology and cannot leverage node features. PyTorch Geometric Temporal [46] implements discrete-time algorithms for spatiotemporal signal processing and is mainly applicable for nodes with aligned historical observations. TGL [74] trains on large-scale dynamic graphs with some engineering tricks. Though TGL has integrated some continuous-time methods, they are somewhat out-of-date, resulting in the lack of state-of-the-art models. Moreover, TGL is implemented by both PyTorch and C++, which needs additional compilation and increases the usage difficulty. In this paper, we present a unified continuous-time dynamic graph learning library with thorough baselines, diverse datasets, extensible implementations, and comprehensive evaluations to facilitate dynamic graph learning research.

# 3 Preliminaries

**Definition 1.** *Dynamic Graph*. *We represent a dynamic graph as a sequence of non-decreasing chronological interactions $\mathcal{G} = \{(u_1, v_1, t_1), (u_2, v_2, t_2), \cdots\}$ with $0 \leq t_1 \leq t_2 \leq \cdots$, where $u_i, v_i \in \mathcal{N}$ denote the source node and destination node of the $i$-th link at timestamp $t_i$. $\mathcal{N}$ is the set of all the nodes. Each node $u \in \mathcal{N}$ can be associated with node feature $\boldsymbol{x}_u \in \mathbb{R}^{d_N}$, and each interaction $(u, v, t)$ has link feature $\boldsymbol{e}_{u,v}^t \in \mathbb{R}^{d_E}$. $d_N$ and $d_E$ denote the dimensions of the node feature and link feature. If the graph is non-attributed, we simply set the node feature and link feature to zero vectors, i.e., $\boldsymbol{x}_u = \mathbf{0}$ and $\boldsymbol{e}_{u,v}^t = \mathbf{0}$.*

**Definition 2.** *Problem Formalization*. *Given the source node $u$, destination node $v$, timestamp $t$, and historical interactions before $t$, i.e., $\{(u', v', t') | t' < t\}$, representation learning on dynamic graph aims to design a model to learn time-aware representations $\boldsymbol{h}_u^t \in \mathbb{R}^d$ and $\boldsymbol{h}_v^t \in \mathbb{R}^d$ for $u$ and $v$ with $d$ as the dimension. We validate the effectiveness of the learned representations via two common tasks in dynamic graph learning: (i) dynamic link prediction, which predicts whether $u$ and $v$ are connected at $t$; (ii) dynamic node classification, which infers the state of $u$ or $v$ at $t$.*

# 4 New Architecture and Unified Library

## 4.1 DyGFormer: Transformer-based Architecture for Dynamic Graph Learning

The framework of our DyGFormer is shown in Figure 1, which employs Transformer [56] as the backbone. Given an interaction $(u, v, t)$, we first extract historical first-hop interactions of source node $u$ and destination node $v$ before timestamp $t$ and obtain two interaction sequences $\mathcal{S}_u^t$ and $\mathcal{S}_v^t$. Next, in addition to computing the encodings of neighbors, links, and time intervals for each sequence, we also encode the frequencies of every neighbor's appearances in both $\mathcal{S}_u^t$ and $\mathcal{S}_v^t$ to exploit the correlations between $u$ and $v$, resulting in four encoding sequences for $u/v$ in total. Then, we divide each encoding sequence into multiple patches and feed all the patches into a Transformer for capturing long-term temporal dependencies. Finally, the outputs of the Transformer are averaged to derive time-aware representations of $u$ and $v$ at timestamp $t$ (i.e., $\boldsymbol{h}_u^t$ and $\boldsymbol{h}_v^t$), which can be applied in various downstream tasks like dynamic link prediction and dynamic node classification.

**Learning from Historical First-hop Interactions**. Unlike most previous methods that require nodes' historical interactions from multiple hops (e.g., DyRep [55], TGAT [64], TGN [45], CAWN [60]), we only learn from the sequences of nodes' historical first-hop interactions, turning the dynamic graph learning task into a simpler sequence learning problem. Mathematically, given an interaction $(u, v, t)$, for source node $u$ and destination node $v$, we obtain the sequences that involve first-hop interactions of $u$ and $v$ before timestamp $t$, which are denoted by $\mathcal{S}_u^t = \{(u, u', t') | t' < t\} \cup \{(u', u, t') | t' < t\}$ and $\mathcal{S}_v^t = \{(v, v', t') | t' < t\} \cup \{(v', v, t') | t' < t\}$, respectively.

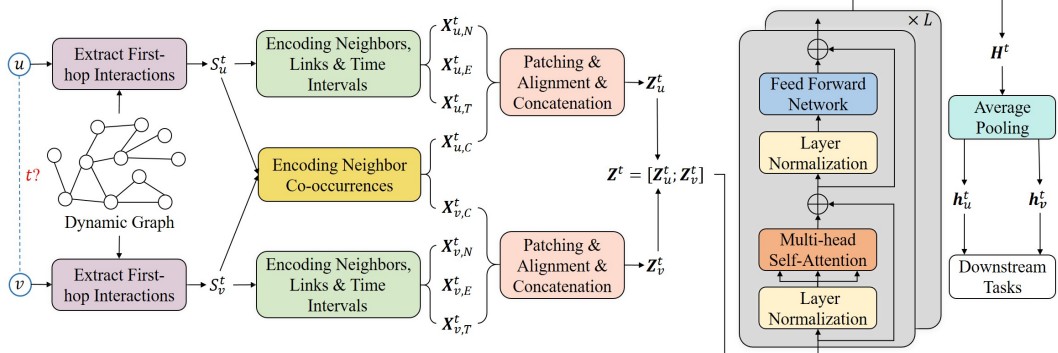

Figure 1: Framework of the proposed model.

**Encoding Neighbors, Links, and Time Intervals**. For source node $u$, we retrieve the features of involved neighbors and links in sequence $\mathcal{S}_u^t$ based on the given features to represent their encodings, which are denoted by $\boldsymbol{X}_{u,N}^t \in \mathbb{R}^{|\mathcal{S}_u^t| \times d_N}$ and $\boldsymbol{X}_{u,E}^t \in \mathbb{R}^{|\mathcal{S}_u^t| \times d_E}$. Following [64], we learn the periodic temporal patterns by encoding the time interval $\Delta t' = t - t'$ via $\sqrt{\frac{1}{d_T}}\left[\cos\left(w_1 \Delta t'\right), \sin\left(w_1 \Delta t'\right), \cdots, \cos\left(w_{d_T} \Delta t'\right), \sin\left(w_{d_T} \Delta t'\right)\right]$, where $w_1, \cdots, w_{d_T}$ are trainable parameters. $d_T$ is the encoding dimension. The time interval encodings of interactions in $\mathcal{S}_u^t$ is denoted by $\boldsymbol{X}_{u,T}^t \in \mathbb{R}^{|\mathcal{S}_u^t| \times d_T}$. We use the same process to get the corresponding encodings for destination node $v$, i.e., $\boldsymbol{X}_{v,N}^t \in \mathbb{R}^{|\mathcal{S}_v^t| \times d_N}$, $\boldsymbol{X}_{v,E}^t \in \mathbb{R}^{|\mathcal{S}_v^t| \times d_E}$, and $\boldsymbol{X}_{v,T}^t \in \mathbb{R}^{|\mathcal{S}_v^t| \times d_T}$.

**Neighbor Co-occurrence Encoding Scheme**. Existing methods separately compute representations of node $u$ and $v$ without modeling their correlations. We present a neighbor co-occurrence encoding scheme to tackle this issue, which assumes the appearing frequency of a neighbor in a sequence indicates its importance, and the occurrences of a neighbor in sequences of $u$ and $v$ (i.e., co-occurrence) could reflect the correlations between $u$ and $v$. That is to say, if $u$ and $v$ have more common historical neighbors in their sequences, they are more likely to interact in the future.

Formally, for each neighbor in the interaction sequence $\mathcal{S}_u^t$ and $\mathcal{S}_v^t$, we count its occurrences in both $\mathcal{S}_u^t$ and $\mathcal{S}_v^t$, and derive a two-dimensional vector. By packing the vectors of all the neighbors together, we can get the neighbor co-occurrence features for $u$ and $v$, which are represented by $\boldsymbol{C}_u^t \in \mathbb{R}^{|\mathcal{S}_u^t| \times 2}$ and $\boldsymbol{C}_v^t \in \mathbb{R}^{|\mathcal{S}_v^t| \times 2}$. For example, suppose the historical neighbors of $u$ and $v$ are $\{a, b, a\}$ and $\{b, b, a, c\}$. The appearing frequencies of $a$, $b$, and $c$ in $u/v$'s historical interactions are $2/1$, $1/2$, and $0/1$, respectively. Therefore, the neighbor co-occurrence features of $u$ and $v$ are denoted by $\boldsymbol{C}_u^t = [[2, 1], [1, 2], [2, 1]]^\top$ and $\boldsymbol{C}_v^t = [[1, 2], [1, 2], [2, 1], [0, 1]]^\top$. Then, we apply a function $f(\cdot)$ to encode the neighbor co-occurrence features by

$$\boldsymbol{X}_{*,C}^t = f\left(\boldsymbol{C}_*^t[:, 0]\right) + f\left(\boldsymbol{C}_*^t[:, 1]\right) \in \mathbb{R}^{|\mathcal{S}_*^t| \times d_C}, \tag{1}$$

where $*$ could be $u$ or $v$. The input and output dimensions of $f(\cdot)$ are 1 and $d_C$. In this paper, we implement $f(\cdot)$ by a two-layer perceptron with ReLU activation [37]. It is important to note that the neighbor co-occurrence encoding scheme is general and can be easily integrated into some dynamic graph learning methods for better results. We will demonstrate its generalizability in Section 5.3.

**Patching Technique**. Instead of focusing on the interaction level, we divide the encoding sequence into multiple non-overlapping patches to break through the bottleneck of existing methods in capturing long-term temporal dependencies. Let $P$ denote the patch size. Each patch is composed of $P$ temporally adjacent interactions with flattened encodings and can preserve local temporal proximities. Take the patching of $\boldsymbol{X}_{u,N}^t \in \mathbb{R}^{|\mathcal{S}_u^t| \times d_N}$ as an example. $\boldsymbol{X}_{u,N}^t$ will be divided into $l_u^t = \lceil \frac{|\mathcal{S}_u^t|}{P} \rceil$ patches in total (note that we will pad $\boldsymbol{X}_{u,N}^t$ if its length $|\mathcal{S}_u^t|$ cannot be divided by $P$), and the patched encoding is represented by $\boldsymbol{M}_{u,N}^t \in \mathbb{R}^{l_u^t \times d_N \cdot P}$. Similarly, we can also get the patched encodings $\boldsymbol{M}_{u,E}^t \in \mathbb{R}^{l_u^t \times d_E \cdot P}$, $\boldsymbol{M}_{u,T}^t \in \mathbb{R}^{l_u^t \times d_T \cdot P}$, $\boldsymbol{M}_{u,C}^t \in \mathbb{R}^{l_u^t \times d_C \cdot P}$, $\boldsymbol{M}_{v,N}^t \in \mathbb{R}^{l_v^t \times d_N \cdot P}$, $\boldsymbol{M}_{v,E}^t \in \mathbb{R}^{l_v^t \times d_E \cdot P}$, $\boldsymbol{M}_{v,T}^t \in \mathbb{R}^{l_v^t \times d_T \cdot P}$, and $\boldsymbol{M}_{v,C}^t \in \mathbb{R}^{l_v^t \times d_C \cdot P}$. Note that when $|\mathcal{S}_u^t|$ becomes

longer, we will correspondingly increase $P$, making the number of patches (i.e., $l_u^t$ and $l_v^t$) at a constant level to reduce the computational cost.

**Transformer Encoder**. We first align the patched encodings to the same dimension $d$ with trainable weight $\boldsymbol{W}_* \in \mathbb{R}^{d_* \cdot P \times d}$ and $\boldsymbol{b}_* \in \mathbb{R}^d$ to obtain $\boldsymbol{Z}_{u,*}^t \in \mathbb{R}^{l_u^t \times d}$ and $\boldsymbol{Z}_{v,*}^t \in \mathbb{R}^{l_v^t \times d}$, where $*$ could be $N, E, T$ or $C$. To be specific, the alignments are realized by

$$\boldsymbol{Z}_{u,*}^t = \boldsymbol{M}_{u,*}^t \boldsymbol{W}_* + \boldsymbol{b}_* \in \mathbb{R}^{l_u^t \times d}, \boldsymbol{Z}_{v,*}^t = \boldsymbol{M}_{v,*}^t \boldsymbol{W}_* + \boldsymbol{b}_* \in \mathbb{R}^{l_v^t \times d}. \tag{2}$$

Then, we concatenate the aligned encodings of $u$ and $v$, and get $\boldsymbol{Z}_u^t = \boldsymbol{Z}_{u,N}^t \| \boldsymbol{Z}_{u,E}^t \| \boldsymbol{Z}_{u,T}^t \| \boldsymbol{Z}_{u,C}^t \in \mathbb{R}^{l_u^t \times 4d}$ and $\boldsymbol{Z}_v^t = \boldsymbol{Z}_{v,N}^t \| \boldsymbol{Z}_{v,E}^t \| \boldsymbol{Z}_{v,T}^t \| \boldsymbol{Z}_{v,C}^t \in \mathbb{R}^{l_v^t \times 4d}$.

Next, we employ a Transformer encoder to capture the temporal dependencies, which is built by stacking $L$ Multi-head Self-Attention (MSA) and Feed-Forward Network (FFN) blocks. The residual connection [20] is employed after every block. We follow [14] by using GELU [21] instead of ReLU [37] between the two-layer perception in each FFN block and applying Layer Normalization (LN) [3] before each block rather than after. Instead of individually processing $\boldsymbol{Z}_u^t$ and $\boldsymbol{Z}_v^t$, our Transformer encoder takes the stacked $\boldsymbol{Z}^t = [\boldsymbol{Z}_u^t; \boldsymbol{Z}_v^t] \in \mathbb{R}^{(l_u^t + l_v^t) \times 4d}$ as inputs, aiming to learn the temporal dependencies within and across the sequences of $u$ and $v$. The calculation process is

$$\text{Attention}(\boldsymbol{Q}, \boldsymbol{K}, \boldsymbol{V}) = \text{Softmax}\left(\frac{\boldsymbol{Q}\boldsymbol{K}^\top}{\sqrt{d_k}}\right)\boldsymbol{V}, \tag{3}$$

$$\text{FFN}(\boldsymbol{O}, \boldsymbol{W}_1, \boldsymbol{b}_1, \boldsymbol{W}_2, \boldsymbol{b}_2) = \text{GELU}(\boldsymbol{O}\boldsymbol{W}_1 + \boldsymbol{b}_1)\boldsymbol{W}_2 + \boldsymbol{b}_2, \tag{4}$$

$$\boldsymbol{O}_i^{t,l} = \text{Attention}\left(\text{LN}(\boldsymbol{Z}^{t,l-1})\boldsymbol{W}_{Q,i}^l, \text{LN}(\boldsymbol{Z}^{t,l-1})\boldsymbol{W}_{K,i}^l, \text{LN}(\boldsymbol{Z}^{t,l-1})\boldsymbol{W}_{V,i}^l\right), \tag{5}$$

$$\boldsymbol{O}^{t,l} = \text{MSA}\left(\boldsymbol{Z}^{t,l-1}\right) + \boldsymbol{Z}^{t,l-1} = \left(\boldsymbol{O}_1^{t,l} \| \cdots \| \boldsymbol{O}_I^{t,l}\right)\boldsymbol{W}_O^l + \boldsymbol{Z}^{t,l-1}, \tag{6}$$

$$\boldsymbol{Z}^{t,l} = \text{FFN}\left(\text{LN}\left(\boldsymbol{O}^{t,l}\right), \boldsymbol{W}_1^l, \boldsymbol{b}_1^l, \boldsymbol{W}_2^l, \boldsymbol{b}_2^l\right) + \boldsymbol{O}^{t,l}. \tag{7}$$

$\boldsymbol{W}_{Q,i}^l \in \mathbb{R}^{4d \times d_k}$, $\boldsymbol{W}_{K,i}^l \in \mathbb{R}^{4d \times d_k}$, $\boldsymbol{W}_{V,i}^l \in \mathbb{R}^{4d \times d_v}$, $\boldsymbol{W}_O^l \in \mathbb{R}^{I \cdot d_v \times 4d}$, $\boldsymbol{W}_1^l \in \mathbb{R}^{4d \times 16d}$, $\boldsymbol{b}_1^l \in \mathbb{R}^{16d}$, $\boldsymbol{W}_2^l \in \mathbb{R}^{16d \times 4d}$ and $\boldsymbol{b}_2^l \in \mathbb{R}^{4d}$ are trainable parameters at the $l$-th layer. We set $d_k = d_v = 4d/I$ with $I$ as the number of attention heads. The input of the first layer is $\boldsymbol{Z}^{t,0} = \boldsymbol{Z}^t \in \mathbb{R}^{(l_u^t + l_v^t) \times 4d}$, and the output of the $L$-th layer is denoted by $\boldsymbol{H}^t = \boldsymbol{Z}^{t,L} \in \mathbb{R}^{(l_u^t + l_v^t) \times 4d}$.

**Time-aware Node Representation**. The time-aware representations of node $u$ and $v$ at timestamp $t$ are derived by averaging their related representations in $\boldsymbol{H}^t$ with an output layer,

$$\begin{aligned}
\boldsymbol{h}_u^t &= \text{MEAN}\left(\boldsymbol{H}^t[: l_u^t, :]\right)\boldsymbol{W}_{out} + \boldsymbol{b}_{out} \in \mathbb{R}^{d_{out}}, \\
\boldsymbol{h}_v^t &= \text{MEAN}\left(\boldsymbol{H}^t[l_u^t : l_u^t + l_v^t, :]\right)\boldsymbol{W}_{out} + \boldsymbol{b}_{out} \in \mathbb{R}^{d_{out}},
\end{aligned} \tag{8}$$

where $\boldsymbol{W}_{out} \in \mathbb{R}^{4d \times d_{out}}$ and $\boldsymbol{b}_{out} \in \mathbb{R}^{d_{out}}$ are trainable weights with $d_{out}$ as the output dimension.

## 4.2 DyGLib: Unified Library for Continuous-Time Dynamic Graph Learning

We introduce a unified library with standard training pipelines, extensible coding interfaces, and comprehensive evaluating strategies for reproducible, scalable, and credible continuous-time dynamic graph learning research. The overall procedure of DyGLib is shown in Figure 3 in Section A.3.

**Standard Training Pipelines**. To eliminate the influence of different training pipelines in previous studies, we unify the data format, create a customized data loader, and train all the methods with the same model trainers. Our standard training pipelines guarantee reproducible performance and enable users to quickly identify the key components of different models. Researchers only need to focus on designing the model architecture without considering other irrelevant implementation details.

**Extensible Coding Interfaces**. We provide extensible coding interfaces for the datasets and algorithms, which are all implemented by PyTorch. These scalable designs enable users to incorporate new datasets and popular models based on their specific requirements, which can significantly reduce the usage difficulty for beginners and allow experts to conveniently validate new ideas. Currently, DyGLib has integrated thirteen datasets from various domains and nine continuous-time dynamic graph learning methods. It is worth noticing that we also found some issues in previous implementations and have fixed them in DyGLib (see details in Section B.3).

**Comprehensive Evaluating Protocols**. DyGLib supports both transductive/inductive dynamic link prediction and dynamic node classification tasks. Most previous works evaluate their methods on the dynamic link prediction task with the random negative sampling strategy but a few models already reach saturation performance under such a strategy, making it hard to distinguish more advanced designs. For more reliable comparisons, we adopt three strategies (i.e., random, historical, and inductive negative sampling strategies) in [44] to comprehensively evaluate the model performance.

## 5 Experiments

In this section, we report the results of various approaches by using DyGLib. We show the superiority of DyGFormer over existing methods and also give an in-depth analysis of DyGFormer.

### 5.1 Experimental Settings

**Datasets and Baselines**. We experiment with thirteen datasets (Wikipedia, Reddit, MOOC, LastFM, Enron, Social Evo., UCI, Flights, Can. Parl., US Legis., UN Trade, UN Vote, and Contact), which are collected by [44] and cover diverse domains. Details of the datasets are shown in Section B.1. We compare DyGFormer with eight popular continuous-time dynamic graph learning baselines that are based on graph convolutions, memory networks, random walks, and sequential models, including JODIE [28], DyRep [55], TGAT [64], TGN [45], CAWN [60], EdgeBank [44], TCL [57], and GraphMixer [12]. We give the descriptions of baselines in Section B.2.

**Evaluation Tasks and Metrics**. We follow [64, 45, 60, 44] to evaluate models for dynamic link prediction, which predicts the probability of a link occurring between two given nodes at a specific time. This task has two settings: the transductive setting aims to predict future links between nodes that are observed during training, and the inductive setting predicts future links between unseen nodes. We use a multi-layer perceptron to take the concatenated representations of two nodes as inputs and return the probability of a link as the output. Average Precision (AP) and Area Under the Receiver Operating Characteristic Curve (AUC-ROC) are adopted as the evaluation metrics. We adopt random (rnd), historical (hist), and inductive (ind) negative sampling strategies in [44] for evaluation, where the latter two strategies are more challenging. Please refer to [44] for more details. We also follow [64, 45] to conduct dynamic node classification, which estimates the state of a node in a given interaction at a specific time. A multi-layer perceptron is employed to map the node representations to the labels. We use AUC-ROC as the evaluation metric due to the label imbalance. For both tasks, we chronologically split each dataset with the ratio of 70%/15%/15% for training/validation/testing.

**Model Configurations**. For baselines, in addition to following their official settings, we also perform an exhaustive grid search to find the optimal configurations of some critical hyperparameters for more reliable comparisons. As DyGFormer can access longer histories, we vary each node's input sequence length from 32 to 4096 by a factor of 2. To keep the computational complexity at a constant level that is irrelevant to the input length, we correspondingly increase the patch size from 1 to 128. Please see Section B.5 for the detailed configurations of different models.

**Implementation Details**. For both tasks, we optimize all models (i.e., excluding EdgeBank which has no trainable parameters) by Adam [27] and use supervised binary cross-entropy loss as the objective function. We train the models for 100 epochs and use the early stopping strategy with a patience of 20. We select the model that achieves the best performance on the validation set for testing. We set the learning rate and batch size to 0.0001 and 200 for all the methods on all the datasets. We run the methods five times with seeds from 0 to 4 and report the average performance to eliminate deviations. Experiments are conducted on an Ubuntu machine equipped with one Intel(R) Xeon(R) Gold 6130 CPU @ 2.10GHz with 16 physical cores. The GPU device is NVIDIA Tesla T4 with 15 GB memory.

### 5.2 Performance Comparisons and Discussions

We report the performance of different methods on the AP metric for transductive dynamic link prediction with three negative sampling strategies in Table 1. The best and second-best results are emphasized by **bold** and underlined fonts. Note that the results are multiplied by 100 for a better display layout. Please refer to Section C.1 and Section C.2 for the results of AP for inductive dynamic link prediction as well as AUC-ROC for transductive and inductive dynamic link prediction tasks.

Table 1: AP for transductive dynamic link prediction with random, historical, and inductive negative sampling strategies. NSS is the abbreviation of Negative Sampling Strategies.

| NSS | Datasets | JODIE | DyRep | TGAT | TGN | CAWN | EdgeBank | TCL | GraphMixer | DyGFormer |
|---|---|---|---|---|---|---|---|---|---|---|
| rnd | Wikipedia | 96.50 ± 0.14 | 94.86 ± 0.06 | 96.94 ± 0.06 | 98.45 ± 0.06 | 98.76 ± 0.03 | 90.37 ± 0.00 | 96.47 ± 0.16 | 97.25 ± 0.03 | **99.03 ± 0.02** |
| | Reddit | 98.31 ± 0.14 | 98.22 ± 0.04 | 98.52 ± 0.02 | 98.63 ± 0.06 | 99.11 ± 0.01 | 94.86 ± 0.00 | 97.53 ± 0.02 | 97.31 ± 0.01 | **99.22 ± 0.01** |
| | MOOC | 80.23 ± 2.44 | 81.97 ± 0.49 | 85.84 ± 0.15 | **89.15 ± 1.60** | 80.15 ± 0.25 | 57.97 ± 0.00 | 82.38 ± 0.24 | 82.78 ± 0.15 | 87.52 ± 0.49 |
| | LastFM | 70.85 ± 2.13 | 71.92 ± 2.21 | 73.42 ± 0.21 | 77.07 ± 3.97 | 86.99 ± 0.06 | 79.29 ± 0.00 | 67.27 ± 2.16 | 75.61 ± 0.24 | **93.00 ± 0.12** |
| | Enron | 84.77 ± 0.30 | 82.38 ± 3.36 | 71.12 ± 0.97 | 86.53 ± 1.11 | 89.56 ± 0.09 | 83.53 ± 0.00 | 79.70 ± 0.71 | 82.25 ± 0.16 | **92.47 ± 0.12** |
| | Social Evo. | 89.89 ± 0.55 | 88.87 ± 0.30 | 93.16 ± 0.17 | 93.57 ± 0.17 | 84.96 ± 0.09 | 74.95 ± 0.00 | 93.13 ± 0.16 | 93.37 ± 0.07 | **94.73 ± 0.01** |
| | UCI | 89.43 ± 1.09 | 65.14 ± 2.30 | 79.63 ± 0.70 | 92.34 ± 1.04 | 95.18 ± 0.06 | 76.20 ± 0.00 | 89.57 ± 1.63 | 93.25 ± 0.57 | **95.79 ± 0.17** |
| | Flights | 95.60 ± 1.73 | 95.29 ± 0.72 | 94.03 ± 0.18 | 97.95 ± 0.14 | 98.51 ± 0.01 | 89.35 ± 0.00 | 91.23 ± 0.02 | 90.99 ± 0.05 | **98.91 ± 0.01** |
| | Can. Parl. | 69.26 ± 0.31 | 66.54 ± 2.76 | 70.73 ± 0.72 | 70.88 ± 2.34 | 69.82 ± 2.34 | 64.55 ± 0.00 | 68.67 ± 2.67 | 77.04 ± 0.46 | **97.36 ± 0.45** |
| | US Legis. | 75.05 ± 1.52 | 75.34 ± 0.39 | 68.52 ± 3.16 | **75.99 ± 0.58** | 70.58 ± 0.48 | 58.39 ± 0.00 | 69.59 ± 0.48 | 70.74 ± 1.02 | 71.11 ± 0.59 |
| | UN Trade | 64.94 ± 0.31 | 63.21 ± 0.93 | 61.47 ± 0.18 | 65.03 ± 1.37 | 65.39 ± 0.12 | 60.41 ± 0.00 | 62.21 ± 0.03 | 62.61 ± 0.27 | **66.46 ± 1.29** |
| | UN Vote | 63.91 ± 0.81 | 62.81 ± 0.80 | 52.21 ± 0.98 | **65.72 ± 2.17** | 52.84 ± 0.10 | 58.49 ± 0.00 | 51.90 ± 0.30 | 52.11 ± 0.16 | 55.55 ± 0.42 |
| | Contact | 95.31 ± 1.33 | 95.98 ± 0.15 | 96.28 ± 0.09 | 96.89 ± 0.56 | 90.26 ± 0.28 | 92.58 ± 0.00 | 92.44 ± 0.12 | 91.92 ± 0.03 | **98.29 ± 0.01** |
| | Avg. Rank | 5.08 | 5.85 | 5.69 | 2.54 | 4.31 | 7.54 | 6.92 | 5.46 | **1.62** |
| hist | Wikipedia | 83.01 ± 0.66 | 79.93 ± 0.56 | 87.38 ± 0.22 | 86.86 ± 0.33 | 71.21 ± 1.67 | 73.35 ± 0.00 | 89.05 ± 0.39 | **90.90 ± 0.10** | 82.23 ± 2.54 |
| | Reddit | 80.03 ± 0.36 | 79.83 ± 0.31 | 79.55 ± 0.20 | 81.22 ± 0.61 | 80.82 ± 0.45 | 73.59 ± 0.00 | 77.14 ± 0.16 | 78.44 ± 0.18 | **81.57 ± 0.67** |
| | MOOC | 78.94 ± 1.25 | 75.60 ± 1.12 | 82.19 ± 0.62 | **87.06 ± 1.93** | 74.05 ± 0.95 | 60.71 ± 0.00 | 77.06 ± 0.41 | 77.77 ± 0.92 | 85.85 ± 0.66 |
| | LastFM | 74.35 ± 3.81 | 74.92 ± 2.46 | 71.59 ± 0.24 | 76.87 ± 4.64 | 69.86 ± 0.43 | 73.03 ± 0.00 | 59.30 ± 2.31 | 72.47 ± 0.49 | **81.57 ± 0.48** |
| | Enron | 69.85 ± 2.70 | 71.19 ± 2.76 | 64.07 ± 1.05 | 73.91 ± 1.76 | 64.73 ± 0.36 | 76.53 ± 0.00 | 70.66 ± 0.39 | **77.98 ± 0.92** | 75.63 ± 0.73 |
| | Social Evo. | 87.44 ± 6.78 | 93.29 ± 0.43 | 95.01 ± 0.44 | 94.45 ± 0.56 | 85.53 ± 0.38 | 80.57 ± 0.00 | 94.74 ± 0.31 | 94.93 ± 0.31 | **97.38 ± 0.14** |
| | UCI | 75.24 ± 5.80 | 55.10 ± 3.14 | 68.27 ± 1.37 | 80.43 ± 2.12 | 65.30 ± 0.43 | 65.50 ± 0.00 | 80.25 ± 2.74 | **84.11 ± 1.35** | 82.17 ± 0.82 |
| | Flights | 66.48 ± 2.59 | 67.61 ± 0.99 | **72.38 ± 0.18** | 66.70 ± 1.64 | 64.72 ± 0.97 | 70.53 ± 0.00 | 70.68 ± 0.24 | 71.47 ± 0.26 | 66.59 ± 0.49 |
| | Can. Parl. | 51.79 ± 0.63 | 63.31 ± 1.23 | 67.13 ± 0.84 | 68.42 ± 3.07 | 66.53 ± 2.77 | 63.84 ± 0.00 | 65.93 ± 3.00 | 74.34 ± 0.87 | **97.00 ± 0.31** |
| | US Legis. | 51.71 ± 5.76 | **86.88 ± 2.25** | 62.14 ± 6.60 | 74.00 ± 7.57 | 68.83 ± 8.23 | 63.22 ± 0.00 | 80.53 ± 3.95 | 81.65 ± 1.02 | 85.30 ± 3.88 |
| | UN Trade | 61.39 ± 1.83 | 59.19 ± 1.07 | 55.74 ± 0.91 | 58.44 ± 5.51 | 55.71 ± 0.38 | **81.32 ± 0.00** | 55.90 ± 1.17 | 57.05 ± 1.22 | 64.41 ± 1.40 |
| | UN Vote | 70.02 ± 0.81 | 69.30 ± 1.12 | 52.96 ± 2.14 | 69.37 ± 3.93 | 51.26 ± 0.04 | **84.89 ± 0.00** | 52.30 ± 2.35 | 51.20 ± 1.60 | 60.84 ± 1.58 |
| | Contact | 95.31 ± 2.13 | 96.39 ± 0.20 | 96.05 ± 0.52 | 93.05 ± 2.35 | 84.16 ± 0.49 | 88.81 ± 0.00 | 93.86 ± 0.21 | 93.36 ± 0.41 | **97.57 ± 0.06** |
| | Avg. Rank | 5.46 | 5.08 | 5.08 | 3.85 | 7.54 | 5.92 | 5.46 | 4.00 | **2.62** |
| ind | Wikipedia | 75.65 ± 0.79 | 70.21 ± 1.58 | 87.00 ± 0.16 | 85.62 ± 0.44 | 74.06 ± 2.62 | 80.63 ± 0.00 | 86.76 ± 0.72 | **88.59 ± 0.17** | 78.29 ± 5.38 |
| | Reddit | 86.98 ± 0.16 | 86.30 ± 0.26 | 89.59 ± 0.24 | 88.10 ± 0.24 | **91.67 ± 0.24** | 85.48 ± 0.00 | 87.45 ± 0.29 | 85.26 ± 0.11 | 91.11 ± 0.40 |
| | MOOC | 65.23 ± 2.19 | 61.66 ± 0.95 | 75.95 ± 0.64 | 77.50 ± 2.91 | 73.51 ± 0.94 | 49.43 ± 0.00 | 74.65 ± 0.54 | 74.27 ± 0.92 | **81.24 ± 0.69** |
| | LastFM | 62.67 ± 4.49 | 64.41 ± 2.70 | 71.13 ± 0.17 | 65.95 ± 5.98 | 67.48 ± 0.77 | **75.49 ± 0.00** | 58.21 ± 0.89 | 68.12 ± 0.33 | 73.97 ± 0.50 |
| | Enron | 68.96 ± 0.98 | 67.79 ± 1.53 | 63.94 ± 1.36 | 70.89 ± 2.72 | 75.15 ± 0.58 | 73.89 ± 0.00 | 71.29 ± 0.32 | 75.01 ± 0.79 | **77.41 ± 0.89** |
| | Social Evo. | 89.82 ± 4.11 | 93.28 ± 0.48 | 94.84 ± 0.44 | 95.13 ± 0.56 | 88.32 ± 0.27 | 83.69 ± 0.00 | 94.90 ± 0.36 | 94.72 ± 0.33 | **97.68 ± 0.10** |
| | UCI | 65.99 ± 1.40 | 54.79 ± 1.76 | 68.67 ± 0.84 | 70.94 ± 0.71 | 64.61 ± 0.48 | 57.43 ± 0.00 | 76.01 ± 1.11 | **80.10 ± 0.51** | 72.25 ± 1.71 |
| | Flights | 69.07 ± 4.02 | 70.57 ± 1.82 | 75.48 ± 0.26 | 71.09 ± 2.72 | 69.18 ± 1.52 | **81.08 ± 0.00** | 74.62 ± 0.18 | 74.87 ± 0.21 | 70.92 ± 1.78 |
| | Can. Parl. | 48.42 ± 0.66 | 58.61 ± 0.86 | 68.82 ± 1.21 | 65.34 ± 2.87 | 67.75 ± 1.00 | 64.74 ± 0.00 | 65.85 ± 1.75 | 69.48 ± 0.63 | **95.44 ± 0.57** |
| | US Legis. | 50.27 ± 5.13 | **83.44 ± 1.16** | 61.91 ± 5.82 | 67.57 ± 6.47 | 65.81 ± 8.52 | 64.74 ± 0.00 | 78.15 ± 3.34 | 79.63 ± 0.84 | 81.25 ± 3.62 |
| | UN Trade | 60.42 ± 1.48 | 60.19 ± 1.24 | 60.61 ± 1.24 | 61.04 ± 6.01 | 62.54 ± 0.67 | **72.97 ± 0.00** | 61.06 ± 1.74 | 60.15 ± 1.29 | 55.79 ± 1.02 |
| | UN Vote | **67.79 ± 1.46** | 67.53 ± 1.98 | 52.89 ± 1.61 | 67.63 ± 2.67 | 52.19 ± 0.34 | 66.30 ± 0.00 | 50.62 ± 0.82 | 51.60 ± 0.73 | 51.91 ± 0.84 |
| | Contact | 93.43 ± 1.78 | 94.18 ± 0.10 | 94.35 ± 0.48 | 90.18 ± 3.28 | 89.31 ± 0.27 | 85.20 ± 0.00 | 91.35 ± 0.21 | 90.87 ± 0.35 | **94.75 ± 0.28** |
| | Avg. Rank | 6.62 | 6.38 | 4.15 | 4.38 | 5.46 | 5.62 | 4.69 | 4.46 | **3.23** |

Since EdgeBank can be only evaluated for transductive dynamic link prediction, we do not show its performance under the inductive setting. From the results, we have two main observations.

Firstly, DyGFormer usually outperforms baselines and achieves an average rank of 2.49/2.69 on AP/AUC-ROC for transductive and 2.69/2.56 for inductive dynamic link prediction across three negative sampling strategies. This is because: (i) The neighbor co-occurrence encoding scheme helps DyGFormer exploit correlations between the source node and destination node, which are often predictive for future links (see Section 5.5). (ii) The patching technique allows DyGFormer to access longer histories and capture long-term temporal dependencies (see Section 5.4). In Table 9 in Section B.5, the input sequence lengths of DyGFormer are much longer than those of baselines on several datasets, indicating that it can utilize longer sequences better. We also observe the varying results of DyGFormer across different negative sampling strategies and give an analysis in Section 5.7.

Secondly, some of our findings of baselines differ from previous reports. For instance, the performance of some baselines can be significantly improved by properly setting some hyperparameters. Additionally, some methods would obtain worse results after we fix the problems or make adaptions in their implementations. More explanations can be found in Section B.4. These observations highlight the importance of rigorously evaluating different methods by a unified library and verify the necessity of introducing DyGLib to facilitate the development of dynamic graph learning.

We also report the results of dynamic node classification in Table 15 in Section C.3. We observe that DyGFormer obtains better performance than most baselines and achieves an impressive average rank of 2.50 among them, demonstrating the superiority of DyGFormer once again.

## 5.3 Generalizability of Neighbor Co-occurrence Encoding Scheme

Our **N**eighbor **Co**-occurrence **E**ncoding scheme (NCoE) is versatile and can be easily integrated with dynamic graph learning methods based on sequential models. Hence, we incorporate NCoE with

Table 2: AP for TCL with NCoE.

| Datasets | TCL | w/ NCoE | Improv. |
|---|---|---|---|
| Wikipedia | 96.47 | 99.09 | 2.72% |
| Reddit | 97.53 | 99.04 | 1.55% |
| MOOC | 82.38 | 86.92 | 5.51% |
| LastFM | 67.27 | 84.02 | 24.90% |
| Enron | 79.70 | 90.18 | 13.15% |
| Social Evo. | 93.13 | 94.06 | 1.00% |
| UCI | 89.57 | 94.69 | 5.72% |
| Flights | 91.23 | 97.71 | 7.10% |
| Can. Parl. | 68.67 | 69.34 | 0.98% |
| US Legis. | 69.59 | 69.47 | -0.17% |
| UN Trade | 62.21 | 63.46 | 2.01% |
| UN Vote | 51.90 | 51.52 | -0.73% |
| Contact | 92.44 | 97.98 | 5.99% |

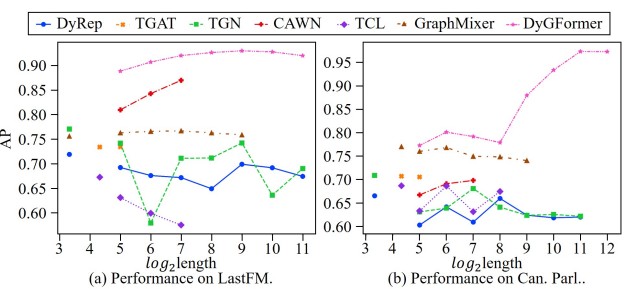

Figure 2: Performance of different methods on LastFM and Can. Parl. with varying input lengths.

TCL and GraphMixer and show their performance in Table 2 and Table 16 in Section C.4. We find TCL and GraphMixer usually yield better results with NCoE, achieving an average improvement of 5.36% and 1.86% over all datasets. This verifies the effectiveness and versatility of the neighbor co-occurrence encoding, and highlights the importance of capturing correlations between nodes. Also, as TCL and DyGFormer are built upon Transformer, TCL w/ NCoE can achieve similar results with DyGFormer on datasets that enjoy shorter input sequences (in which cases the patching technique in DyGFormer contributes little). However, when datasets exhibit more obvious long-term temporal dependencies (e.g., LastFM, Can. Parl.), the performance gaps become more significant.

## 5.4 Advantages of Patching Technique

We validate the advantages of our patching technique in preserving the local temporal proximities and reducing the computational complexity, which helps DyGFormer effectively and efficiently utilize longer histories. We conduct experiments on LastFM and Can. Parl. since they can benefit from longer historical records. For baselines, we sample more neighbors or perform more causal anonymous walks (starting from 32) to make them access longer histories. The results are depicted in Figure 2, where the x-axis is represented by a logarithmic scale with base 2. We also plot the performance of baselines with the optimal length by unconnected points based on Table 9 in Section B.5. Note that the results of some baselines are incomplete since they raise the out-of-memory error when the lengths are longer. For example, TGAT is only computationally feasible when extending the input length to 32, resulting in two discrete points with lengths 20 (the optimal length) and 32.

From Figure 2, we conclude that: (i) most of the baselines perform worse when the input lengths become longer, indicating they lack the ability to capture long-term temporal dependencies; (ii) the baselines usually encounter expensive computational costs when computing on longer histories. Although memory network-based methods (i.e., DyRep and TGN) can handle longer histories with affordable computational costs, they cannot benefit from longer histories due to the potential issues of vanishing or exploding gradients; (iii) DyGFormer consistently achieves gains from longer sequences, demonstrating the advantages of the patching technique in leveraging longer histories.

We also compare the running time and memory usage of DyGFormer with and without the patching technique during the training process. The results are shown in Table 17 in Section C.5. We could observe that the patching technique efficiently reduces model training costs in both time and space, allowing DyGFormer to access longer histories. As the input sequence length increases, the reductions become more significant. Moreover, with the patching technique, we find that DyGFormer achieves an average improvement of 0.31% and 0.74% in performance on LastFM and Can. Parl. than DyGFormer without patching. This observation further demonstrates the advantage of our patching technique in leveraging the local temporal proximities for better results.

## 5.5 Verification of the Motivation of Neighbor Co-occurrence Encoding Scheme

To verify the motivation of NCoE (i.e., nodes with more common historical neighbors tend to interact in the future), we compare the performance of DyGFormer and DyGFormer without NCoE. We choose 0.5 as the threshold and use TP, TN, FN, and FP to denote True/False Positive/Negative.

Common Neighbor Ratio (CNR) is defined as the ratio of common neighbors in source node $u$'s sequence $S_u$ and destination node $v$'s sequence $S_v$, i.e., $|S_u \cap S_v|/|S_u \cup S_v|$. We focus on links whose predictions of DyGFormer w/o NCoE are changed by DyGFormer (i.e., FN→TP, FP→TN, TP→FN, and TN→FP). We define Changed Link Ratio (CLR) as the ratio of the changed links to their original set, which is respectively computed by |FN→TP|/|FN|, |FP→TN|/|FP|, |TP→FN|/|TP|, and |TN→FP|/|TN|. If NCoE is helpful, DyGFormer will revise more wrong predictions (more FN→TP and FP→TN) and make fewer incorrect changes (fewer TP→FN and TN→FP). We report CLR and average CNR of links in the above sets on five typical datasets in Table 3.

Table 3: CLR and CNR of changes made by DyGFormer.

| Datasets | CLR (%) | | | | CNR (%) | | | |
|---|---|---|---|---|---|---|---|---|
| | FN→TP | FP→TN | TP→FN | TN→FP | FN→TP | FP→TN | TP→FN | TN→FP |
| Wikipedia | 68.36 | 72.73 | 1.68 | 1.69 | 18.16 | 0.01 | 0.10 | 2.49 |
| UCI | 71.45 | 94.11 | 7.29 | 1.82 | 19.08 | 2.49 | 3.35 | 13.02 |
| Flights | 83.66 | 83.83 | 1.73 | 2.11 | 37.09 | 2.28 | 7.06 | 20.28 |
| US Legis. | 31.63 | 23.67 | 6.63 | 1.59 | 69.92 | 62.13 | 61.14 | 63.80 |
| UN Vote | 44.02 | 36.46 | 28.95 | 30.53 | 78.57 | 81.39 | 80.86 | 77.02 |

We find NCoE effectively helps DyGFormer rectify wrong predictions of DyGFormer w/o NCoE on datasets with *significantly higher CNR of positive links than negative ones*, which happens with most datasets. Concretely, for Wikipedia, UCI, and Flights, their CNRs of FN→TP are much higher than FP→TN (e.g., 37.09% vs. 2.28% on Flights) and DyGFormer revises most wrong predictions of DyGFormer w/o NCoE (e.g., 83.66% for positive links in FN and 83.83% for negative links in FP on Flights). Corrections made by our encoding scheme are less obvious on datasets whose *CNRs between positive and negative links are similar*, which occurs in only 2 of 13 datasets. For US Legis. and UN Vote, their CNRs between FN and FP are analogous (e.g., 69.92% vs. 62.13% on US Legis.), weakening the advantage of our neighbor co-occurrence encoding scheme (e.g., only 31.63%/23.67% of positive/negative links are corrected in FN/FP on US Legis.). Therefore, we conclude that the neighbor co-occurrence encoding scheme helps DyGFormer capture common historical neighbors in $S_u$ and $S_v$, and bring better results in most cases.

## 5.6 When Will DyGFormer Be a Good Choice?

Note that DyGFormer is superior to baselines by 1) exploring the source and destination nodes' correlations from their historical sequences by neighbor co-occurrence encoding scheme; 2) using the patching technique to attend longer histories. Thus, DyGFormer tends to perform better on datasets that favor these two designs. We define Link Ratio (LR) as the ratio of links in their corresponding positive or negative set, which can be computed by TP/(TP+FN), TN/(TN+FP), FN/(TP+FN), and FP/(TN+FP). As a method with more TP and TN (i.e., fewer FN and FP) is better, we report the results of LR and average CNR of links in TP and TN on five typical datasets in Table 4.

Table 4: LR and CNR of TP and TN with random negative sampling strategy.

| Datasets | LR (%) | | CNR (%) | |
|---|---|---|---|---|
| | TP | TN | TP | TN |
| Wikipedia | 92.74 | 97.19 | 59.09 | 0.01 |
| UCI | 82.70 | 96.77 | 28.03 | 1.45 |
| Flights | 96.13 | 95.33 | 47.58 | 1.40 |
| US Legis. | 78.95 | 56.83 | 75.18 | 53.98 |
| UN Vote | 65.18 | 45.43 | 56.24 | 76.02 |

Table 5: LR and CNR of FP under random, historical, and inductive negative sampling strategy.

| Datasets | LR(%) | | | CNR(%) | | |
|---|---|---|---|---|---|---|
| | rnd | hist | ind | rnd | hist | ind |
| Wikipedia | 2.81 | 89.28 | 94.53 | 0.02 | 14.00 | 11.66 |
| UCI | 3.23 | 64.93 | 76.42 | 9.98 | 12.22 | 13.81 |
| Flights | 4.67 | 94.52 | 92.94 | 0.01 | 35.62 | 30.29 |
| US Legis. | 43.17 | 17.31 | 21.21 | 79.40 | 87.51 | 75.51 |
| UN Vote | 54.57 | 39.90 | 52.92 | 79.60 | 75.53 | 79.15 |

We observe when *CNR of TP is significantly higher than CNR of TN in the datasets, DyGFormer often outperforms baselines* (most datasets satisfy this property). For Wikipedia, UCI, and Flights, their CNRs of TP are much higher than those of TN (e.g., 59.09% vs. 0.01% on Wikipedia). Such a characteristic matches the motivation of our neighbor co-occurrence encoding scheme, enabling DyGFormer to correctly predict most links (e.g., 92.74% of positive links and 97.19% of negative links are properly predicted on Wikipedia). Moreover, as LastFM and Can. Parl. can gain from

longer histories (see Figure 2 and Table 9 in Section B.5), DyGFormer is significantly better than baselines on these two datasets. When *CNRs of TP and TN are less distinguishable in the datasets, DyGFormer may perform worse* (only 2 out of 13 datasets show this property). For US Legis., the CNRs between TP and TN are close (i.e., 75.18% vs. 53.98%), making DyGFormer worse than memory-based baselines (i.e., JODIE, DyRep, and TGN). For UN Vote, its CNR of TP is even lower than that of TN (i.e., 56.24% vs. 76.02%), which is opposite to our motivation, leading to poor results of DyGFormer than a few baselines. Since these two datasets cannot obviously gain from longer sequences either (see Table 9 in Section B.5), DyGFormer obtains worse results on them. Thus, we conclude that for datasets with much higher CNR of TP than CNR of TN or datasets that can benefit from longer histories, DyGFormer is a good choice. Otherwise, we may need to try other methods.

### 5.7 Why do DyGFormer's Performance Vary across Different Negative Sampling Strategies?

Compared with the random (rnd) strategy, historical (hist) and inductive (ind) strategies will sample previous links as negative ones. This makes previous positive links negative, which may hurt the performance DyGFormer since the motivation of our neighbor co-occurrence encoding scheme is violated. As positive links are identical among rnd, hist, and ind, we compute LR and the average CNR of links in FP and show results in Table 5.

We find *when hist or ind causes several magnitudes higher CNR of FP than rnd in the datasets, DyGFormer drops sharply*. For Wikipedia, UCI, and Flights, the CNRs of FP with hist/ind are much higher than rnd (e.g., 14.00%/11.66% vs. 0.02% on Wikipedia). This misleads DyGFormer to predict negative links as positive and causes drops (e.g., 89.28%/94.53% of negative links are incorrectly predicted with hist/ind on Wikipedia, while only 2.81% are wrong with rnd). We also note the drops in UCI are milder since the changes in CNR caused by hist or ind vs. rnd are less obvious than changes in Wikipedia and Flights. *When changes in CNR of FP caused by hist or ind are not obvious in the datasets, DyGFormer is less affected.* Since hist/ind makes little changes in CNRs of FP on US Legis., we find it ranks second with hist/ind, which may indicate DyGFormer is less influenced by the neighbor co-occurrence encoding scheme and generalizes well to various negative sampling strategies. For UN Vote, although its CNRs of FP are not affected by hist and ind either, DyGFormer still performs worse due to its inferior performance with rnd. Hence, we deduce that our neighbor co-occurrence encoding may be sometimes fragile to various negative sampling strategies if its motivation is violated, leading to the varying performance of DyGFormer.

### 5.8 Ablation Study

Finally, we validate the effectiveness of the neighbor co-occurrence encoding, time encoding, and mixing of the sequence of source node and destination node in DyGFormer. From Figure 4 in Section C.6, we observe that DyGFormer obtains the best performance when using all the components, and the results would be worse when any component is removed. This illustrates the necessity of each design in DyGFormer. Please refer to Section C.6 for detailed implementations and discussions.

## 6    Conclusion

In this paper, we proposed a new Transformer-based architecture (DyGFormer) and a unified library (DyGLib) to foster the development of dynamic graph learning. DyGFormer differs from previous methods in (i) a neighbor co-occurrence encoding scheme to exploit the correlations of nodes in each interaction; and (ii) a patching technique to help the model capture long-term temporal dependencies. DyGLib served as a toolkit for reproducible, scalable, and credible continuous-time dynamic graph learning with standard training pipelines, extensible coding interfaces, and comprehensive evaluating protocols. We hope our work can provide new perspectives on designing new dynamic graph learning frameworks and encourage more researchers to dive into this field. In the future, we will continue to enrich DyGLib by incorporating the recently released datasets and state-of-the-art models.

## Acknowledgments and Disclosure of Funding

This work was supported by the National Natural Science Foundation of China (No. 62272023 and 51991395) and the Fundamental Research Funds for the Central Universities (No. YWF-23-L-1203).

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
