# A    Additional Discussions and Details

## A.1    Limitations

One potential limitation of DyGFormer lies in the ignorance of high-order relationships between nodes since it solely learns from the first-hop interactions of nodes. In certain scenarios where nodes' high-order relationships are essential, DyGFormer may be suboptimal compared with baselines that learn the higher-order interactions. However, trivially feeding the multi-hop neighbors of nodes into DyGFormer would incur expensive computational costs. It is promising to design more efficient and effective frameworks to model nodes' high-order relationships for dynamic graph learning.

Another potential limitation is the sensitivity of the neighbor co-occurrence encoding scheme against different negative sampling strategies (discussed in Section 5.7). When the assumption of our neighbor co-occurrence encoding scheme is violated, its performance may drop drastically in some cases. It is an insightful direction to design more robust encoding schemes to tackle this issue.

## A.2    Licenses

All the used codes and datasets are publicly available and permit usage for research purposes under either MIT License or Apache License 2.0.

## A.3    Overall Procedure of DyGLib

Figure 3 shows the overall procedure of DyGLib.

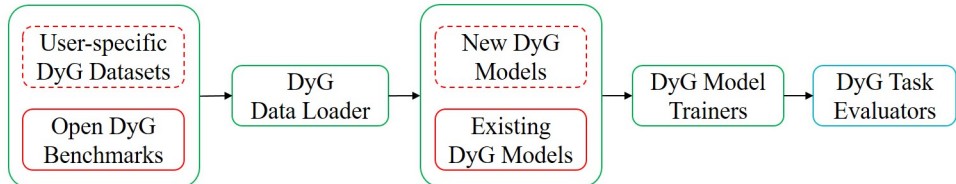

Figure 3: DyGLib is equipped with standard training pipelines, extensible coding interfaces, and comprehensive evaluating protocols. DyG denotes the abbreviation of Dynamic Graph.

# B    Detailed Experimental Settings

## B.1    Descriptions of Datasets

We use thirteen datasets collected by [44] in the experiments, which are publicly available[2]:

- **Wikipedia** is a bipartite interaction graph that contains the edits on Wikipedia pages over a month. Nodes represent users and pages, and links denote the editing behaviors with timestamps. Each link is associated with a 172-dimensional Linguistic Inquiry and Word Count (LIWC) feature [42]. This dataset additionally contains dynamic labels that indicate whether users are temporarily banned from editing.

- **Reddit** is bipartite and records the posts of users under subreddits during one month. Users and subreddits are nodes, and links are the timestamped posting requests. Each link has a 172-dimensional LIWC feature. This dataset also includes dynamic labels representing whether users are banned from posting.

- **MOOC** is a bipartite interaction network of online sources, where nodes are students and course content units (e.g., videos and problem sets). Each link denotes a student's access behavior to a specific content unit and is assigned with a 4-dimensional feature.

- **LastFM** is bipartite and consists of the information about which songs were listened to by which users over one month. Users and songs are nodes, and links denote the listening behaviors of users.

---

[2]https://zenodo.org/record/7213796#.Y1cO6y8r30o

- **Enron** records the email communications between employees of the ENRON energy corporation over three years.

- **Social Evo.** is a mobile phone proximity network that monitors the daily activities of an entire undergraduate dormitory for a period of eight months, where each link has a 2-dimensional feature.

- **UCI** is an online communication network, where nodes are university students and links are messages posted by students.

- **Flights** is a dynamic flight network that displays the development of air traffic during the COVID-19 pandemic. Airports are represented by nodes and the tracked flights are denoted as links. Each link is associated with a weight, indicating the number of flights between two airports in a day.

- **Can. Parl.** is a dynamic political network that records the interactions between Canadian Members of Parliament (MPs) from 2006 to 2019. Each node represents an MP from an electoral district and a link is created when two MPs both vote "yes" on a bill. The weight of each link refers to the number of times that one MP voted "yes" for another MP in a year.

- **US Legis.** is a senate co-sponsorship network that tracks social interactions between legislators in the US Senate. The weight of each link specifies the number of times two congresspersons have co-sponsored a bill in a given congress.

- **UN Trade** contains the food and agriculture trade between 181 nations for more than 30 years. The weight of each link indicates the total sum of normalized agriculture import or export values between two particular countries.

- **UN Vote** records roll-call votes in the United Nations General Assembly. If two nations both voted "yes" to an item, the weight of the link between them is increased by one.

- **Contact** describes how the physical proximity evolves among about 700 university students over a month. Each student has a unique identifier and links denote that they are within close proximity to each other. Each link is associated with a weight, revealing the physical proximity between students.

We show the statistics of the datasets in Table 6, where #N&L Feat stands for the dimensions of node and link features. We notice a slight difference between the statistics of the Contact dataset reported in [44] (which has 694 nodes and 2,426,280 links) and our own calculations, although both of them are computed based on the released dataset by [44]. We ultimately report our statistics of the Contact dataset in this paper.

Table 6: Statistics of the datasets.

| Datasets | Domains | #Nodes | #Links | #N&L Feat | Bipartite | Duration | Unique Steps | Time Granularity |
|---|---|---|---|---|---|---|---|---|
| Wikipedia | Social | 9,227 | 157,474 | – & 172 | True | 1 month | 152,757 | Unix timestamps |
| Reddit | Social | 10,984 | 672,447 | – & 172 | True | 1 month | 669,065 | Unix timestamps |
| MOOC | Interaction | 7,144 | 411,749 | – & 4 | True | 17 months | 345,600 | Unix timestamps |
| LastFM | Interaction | 1,980 | 1,293,103 | – & – | True | 1 month | 1,283,614 | Unix timestamps |
| Enron | Social | 184 | 125,235 | – & – | False | 3 years | 22,632 | Unix timestamps |
| Social Evo. | Proximity | 74 | 2,099,519 | – & 2 | False | 8 months | 565,932 | Unix timestamps |
| UCI | Social | 1,899 | 59,835 | – & – | False | 196 days | 58,911 | Unix timestamps |
| Flights | Transport | 13,169 | 1,927,145 | – & 1 | False | 4 months | 122 | days |
| Can. Parl. | Politics | 734 | 74,478 | – & 1 | False | 14 years | 14 | years |
| US Legis. | Politics | 225 | 60,396 | – & 1 | False | 12 congresses | 12 | congresses |
| UN Trade | Economics | 255 | 507,497 | – & 1 | False | 32 years | 32 | years |
| UN Vote | Politics | 201 | 1,035,742 | – & 1 | False | 72 years | 72 | years |
| Contact | Proximity | 692 | 2,426,279 | – & 1 | False | 1 month | 8,064 | 5 minutes |

### B.2 Descriptions of Baselines

We select the following eight baselines:

- **JODIE** is designed for temporal bipartite networks of user-item interactions. It employs two coupled recurrent neural networks to update the states of users and items. A projection operation is introduced to learn the future representation trajectory of each user/item [28].

- **DyRep** proposes a recurrent architecture to update node states upon each interaction. It also includes a temporal-attentive aggregation module to consider the temporally evolving structural information in dynamic graphs [55].

- **TGAT** computes the node representation by aggregating features from each node's temporal-topological neighbors based on the self-attention mechanism. It is also equipped with a time encoding function for capturing temporal patterns [64].

- **TGN** maintains an evolving memory for each node and updates this memory when the node is observed in an interaction, which is achieved by the message function, message aggregator, and memory updater. An embedding module is leveraged to generate the temporal representations of nodes [45].

- **CAWN** first extracts multiple causal anonymous walks for each node, which can explore the causality of network dynamics and generate relative node identities. Then, it utilizes recurrent neural networks to encode each walk and aggregates these walks to obtain the final node representation [60].

- **EdgeBank** is a pure memory-based approach without trainable parameters for transductive dynamic link prediction. It stores the observed interactions in the memory unit and updates the memory through various strategies. An interaction will be predicted as positive if it was retained in the memory, and negative otherwise [44]. The publication presents two updating strategies, but their official implementations include two more[3]. To be specific, the four variants of EdgeBank are: $EdgeBank_\infty$ with unlimited memory to store all the observed edges; $EdgeBank_{tw\text{-}ts}$ and $EdgeBank_{tw\text{-}re}$, which only remember edges within a fixed-size time window from the immediate past. The window size of $EdgeBank_{tw\text{-}ts}$ is set to the duration of the test split, while $EdgeBank_{tw\text{-}re}$ makes it related to the time intervals of repeated edges; $EdgeBank_{th}$ that retains edges with appearing counts higher than a threshold. We experiment with all four variants and report the best performance among them.

- **TCL** first generates each node's interaction sequence by performing a breadth-first search algorithm on the temporal dependency interaction sub-graph. Then, it presents a graph transformer that considers both graph topology and temporal information to learn node representations. It also incorporates a cross-attention operation for modeling the inter-dependencies between two interaction nodes [57].

- **GraphMixer** shows that a fixed time encoding function performs better than the trainable version. It incorporates the fixed function into a link encoder based on MLP-Mixer [54] to learn from temporal links. A node encoder with neighbor mean-pooing is employed to summarize node features [12].

## B.3 Some Problematic Implementations in Baselines

JODIE, DyRep, and TGN models are based on memory networks, and their implementations have designed the raw messages to avoid information leakage. However, they fail to store the raw messages when saving models because the raw messages are maintained in a dictionary and thus cannot be saved as model parameters[4]. Our DyGLib has addressed this issue by additionally saving the correct raw messages when saving models. In the official implementation of CAWN, the encoding of each causal anonymous walk is represented by the embedding at the last position of the walk[5]. However, some walks are padded to support mini-batch training in practice, making the last position of these padded walks meaningless. To get the correct encoding, it is necessary to use the actual length of each walk. Moreover, there are issues with the nodeedge2idx dictionary computed by the get_ts2idx function[6] if multiple interactions simultaneously occur at the last timestamp. This would lead to information leakage since the model can potentially access more interactions for a given interaction even if they happen at the same time. Our DyGLib has fixed these problems as well.

---

[3]`https://github.com/fpour/DGB/blob/main/EdgeBank/link_pred/edge_bank_baseline.py`

[4]`https://github.com/twitter-research/tgn/blob/2aa295a1f182137a6ad56328b43cb3d8223cae77/train_self_supervised.py#L302`

[5]`https://github.com/snap-stanford/CAW/blob/f994ff2b2c29778e6250b6a9928fd9943e0163f7/module.py#L1069`

[6]`https://github.com/snap-stanford/CAW/blob/f994ff2b2c29778e6250b6a9928fd9943e0163f7/graph.py#L79`

## B.4 Some Inconsistent Observations with Previous Reports

In the experiments, we find the behaviors of baselines are inconsistent with their previous reports in some cases. We provide detailed illustrations and attribute these phenomena to the following reasons.

**Suboptimal Settings of Hyperparameters**. Some important hyperparameters in the baselines are not sufficiently fine-tuned, such as the dropout rate, the number of sampled neighbors, the number of random walks, the length of input sequences, and the neighbor sampling strategies. In this paper, we perform the grid search to find the best settings of these hyperparameters and observe that the performance of many baselines can be significantly improved by properly setting certain hyperparameters. Take TGAT for transductive dynamic link prediction with the random negative sampling strategy as an example (see Table 1). Compared with the performance in [44], the results of AP are significantly improved on datasets like MOOC (from 0.61 to 0.86), LastFM (from 0.50 to 0.73), Enron (from 0.59 to 0.71), Social Evo. (from 0.76 to 0.93), Flights (from 0.89 to 0.94), and Contact (from 0.58 to 0.96). Similar improvements can also be found on JODIE, DyRep, and TGN on several datasets.

**Usage of Problematic Implementations**. Some previous studies utilize problematic implementations in their experiments (illustrated in Section B.3), and the reported results may not be rigorous. Therefore, some methods would obtain worse results after we fix the problems in their implementations. Take CAWN for transductive dynamic link prediction with the random negative sampling strategy as an example (see Table 1). Compared with the results in [44], the performance on the AP metric of CAWN drops sharply on datasets like LastFM (from 0.98 to 0.87), Can. Parl. (from 0.94 to 0.70), US Legis. (from 0.97 to 0.71), UN Trade (from 0.97 to 0.65), and UN Vote (from 0.82 to 0.53). This is because [44] uses the problematic implementations of CAWN to conduct experiments and the results will sometimes become worse after fixing the issues.

**Adaptions for Evaluations**. There also exist some differences between GraphMixer's results in the original paper and our work because we modify its implementation to fit our evaluations. The original GraphMixer could only be evaluated for transductive dynamic link prediction. In this work, we remove the one-hot encoding of nodes in GraphMixer to adapt it to the inductive setting, which may lead to decreased performance in certain situations. Take the performance of GraphMixer for transductive dynamic link prediction with the random negative sampling strategy as an example (see Table 1). Compared with the results in [12], the performance of GraphMixer slightly drops on datasets like Wikipedia (from 0.9985 to 0.9725) and Reddit (from 0.9993 to 0.9731).

## B.5 Configurations of Different Methods

We first present the official settings of baselines as well as the configurations of DyGFormer. These configurations remain unchanged across all the datasets.

- **JODIE**:
    - Dimension of time encoding: 100
    - Dimension of node memory: 172
    - Dimension of output representation: 172
    - Memory updater: vanilla recurrent neural network
- **DyRep**:
    - Dimension of time encoding: 100
    - Dimension of node memory: 172
    - Dimension of output representation: 172
    - Number of graph attention heads: 2
    - Number of graph convolution layers: 1
    - Memory updater: vanilla recurrent neural network
- **TGAT**:
    - Dimension of time encoding: 100
    - Dimension of output representation: 172
    - Number of graph attention heads: 2

- – Number of graph convolution layers: 2
- **TGN**:
  - – Dimension of time encoding: 100
  - – Dimension of node memory: 172
  - – Dimension of output representation: 172
  - – Number of graph attention heads: 2
  - – Number of graph convolution layers: 1
  - – Memory updater: gated recurrent unit [10]
- **CAWN**:
  - – Dimension of time encoding: 100
  - – Dimension of position encoding: 172
  - – Dimension of output representation: 172
  - – Number of attention heads for encoding walks: 8
  - – Length of each walk (including the target node): 2
  - – Time scaling factor $\alpha$: 1e-6
- **TCL**:
  - – Dimension of time encoding: 100
  - – Dimension of depth encoding: 172
  - – Dimension of output representation: 172
  - – Number of attention heads: 2
  - – Number of Transformer layers: 2
- **GraphMixer**:
  - – Dimension of time encoding: 100
  - – Dimension of output representation: 172
  - – Number of MLP-Mixer layers: 2
  - – Time gap $T$: 2000
- **DyGFormer**:
  - – Dimension of time encoding $d_T$: 100
  - – Dimension of neighbor co-occurrence encoding $d_C$: 50
  - – Dimension of aligned encoding $d$: 50
  - – Dimension of output representation $d_{out}$: 172
  - – Number of attention heads $I$: 2
  - – Number of Transformer layers $L$: 2

Then, we perform the grid search to find the best settings of some critical hyperparameters, where the searched ranges and related methods are shown in Table 7. It is worth noticing that DyGFormer can directly handle nodes with sequence lengths shorter than the defined length. When the sequence length exceeds the specified length, we select the most recent interactions up to the defined length.

Finally, we show the hyperparameter settings of various methods that are determined by the grid search in Table 8, Table 9, Table 10 and Table 11.

## C  Detailed Experimental Results

### C.1  Additional Results for Transductive Dynamic Link Prediction

We show the AUC-ROC for transductive dynamic link prediction with three negative sampling strategies in Table 12.

### C.2  Additional Results for Inductive Dynamic Link Prediction

We present the AP and AUC-ROC for inductive dynamic link prediction with three negative sampling strategies in Table 13 and Table 14 .

Table 7: Searched ranges of hyperparameters and the related methods.

| Hyperparameters | Searched Ranges | Related Methods |
|---|---|---|
| Dropout Rate [52] | [0.0, 0.1, 0.2, 0.3, 0.4, 0.5, 0.6] | JODIE, DyRep, TGAT, TGN, CAWN, TCL, GraphMixer, DyGFormer |
| Number of Sampled Neighbors | [10, 20, 30] | DyRep, TGAT, TGN, TCL, GraphMixer |
| Neighbor Sampling Strategies | [uniform, recent] | DyRep, TGAT, TGN, TCL, GraphMixer |
| Number of Causal Anonymous Walks | [16, 32, 64, 128] | CAWN |
| Memory Updating Variants | [EdgeBank$_\infty$, EdgeBank$_{tw-ts}$, EdgeBank$_{tw-re}$, EdgeBank$_{th}$] | EdgeBank |
| Length of Input Sequences & Patch Size | [32 & 1, 64 & 2, 128 & 4, 256 & 8, 512 & 16, 1024 & 32, 2048 & 64, 4096 & 128] | DyGFormer |

Table 8: Configurations of the dropout rate of different methods.

| Datasets | JODIE | DyRep | TGAT | TGN | CAWN | TCL | GraphMixer | DyGFormer |
|---|---|---|---|---|---|---|---|---|
| Wikipedia | 0.1 | 0.1 | 0.1 | 0.1 | 0.1 | 0.1 | 0.5 | 0.1 |
| Reddit | 0.1 | 0.1 | 0.1 | 0.1 | 0.1 | 0.1 | 0.5 | 0.2 |
| MOOC | 0.2 | 0.0 | 0.1 | 0.2 | 0.1 | 0.1 | 0.4 | 0.1 |
| LastFM | 0.3 | 0.0 | 0.1 | 0.3 | 0.1 | 0.1 | 0.0 | 0.1 |
| Enron | 0.1 | 0.0 | 0.2 | 0.0 | 0.1 | 0.1 | 0.5 | 0.0 |
| Social Evo. | 0.1 | 0.1 | 0.1 | 0.0 | 0.1 | 0.0 | 0.3 | 0.1 |
| UCI | 0.4 | 0.0 | 0.1 | 0.1 | 0.1 | 0.0 | 0.4 | 0.1 |
| Flights | 0.1 | 0.1 | 0.1 | 0.1 | 0.1 | 0.1 | 0.2 | 0.1 |
| Can. Parl. | 0.0 | 0.0 | 0.2 | 0.3 | 0.0 | 0.2 | 0.2 | 0.1 |
| US Legis. | 0.2 | 0.0 | 0.1 | 0.1 | 0.1 | 0.3 | 0.4 | 0.0 |
| UN Trade | 0.4 | 0.1 | 0.1 | 0.2 | 0.1 | 0.0 | 0.1 | 0.0 |
| UN Vote | 0.1 | 0.1 | 0.2 | 0.1 | 0.1 | 0.0 | 0.0 | 0.2 |
| Contact | 0.1 | 0.0 | 0.1 | 0.1 | 0.1 | 0.0 | 0.1 | 0.0 |

Table 9: Configurations of the number of sampled neighbors, the number of causal anonymous walks, and the length of input sequences & the patch size of different methods.

| Datasets | DyRep | TGAT | TGN | CAWN | TCL | GraphMixer | DyGFormer |
|---|---|---|---|---|---|---|---|
| Wikipedia | 10 | 20 | 10 | 32 | 20 | 30 | 32 & 1 |
| Reddit | 10 | 20 | 10 | 32 | 20 | 10 | 64 & 2 |
| MOOC | 10 | 20 | 10 | 64 | 20 | 20 | 256 & 8 |
| LastFM | 10 | 20 | 10 | 128 | 20 | 10 | 512 & 16 |
| Enron | 10 | 20 | 10 | 32 | 20 | 20 | 256 & 8 |
| Social Evo. | 10 | 20 | 10 | 64 | 20 | 20 | 32 & 1 |
| UCI | 10 | 20 | 10 | 64 | 20 | 20 | 32 & 1 |
| Flights | 10 | 20 | 10 | 64 | 20 | 20 | 256 & 8 |
| Can. Parl. | 10 | 20 | 10 | 128 | 20 | 20 | 2048 & 64 |
| US Legis. | 10 | 20 | 10 | 32 | 20 | 20 | 256 & 8 |
| UN Trade | 10 | 20 | 10 | 64 | 20 | 20 | 256 & 8 |
| UN Vote | 10 | 20 | 10 | 64 | 20 | 20 | 128 & 4 |
| Contact | 10 | 20 | 10 | 64 | 20 | 20 | 32 & 1 |

Table 10: Configurations of neighbor sampling strategies of different methods.

| Datasets | DyRep | TGAT | TGN | TCL | GraphMixer |
|---|---|---|---|---|---|
| Wikipedia | recent | recent | recent | recent | recent |
| Reddit | recent | uniform | recent | uniform | recent |
| MOOC | recent | recent | recent | recent | recent |
| LastFM | recent | recent | recent | recent | recent |
| Enron | recent | recent | recent | recent | recent |
| Social Evo. | recent | recent | recent | recent | recent |
| UCI | recent | recent | recent | recent | recent |
| Flights | recent | recent | recent | recent | recent |
| Can. Parl. | uniform | uniform | uniform | uniform | uniform |
| US Legis. | recent | recent | recent | uniform | recent |
| UN Trade | recent | uniform | recent | uniform | uniform |
| UN Vote | recent | recent | uniform | uniform | uniform |
| Contact | recent | recent | recent | recent | recent |

Table 11: Configurations of the variants of EdgeBank with three negative sampling strategies.

| Datasets | Random | Historical | Inductive |
|---|---|---|---|
| Wikipedia | $\text{EdgeBank}_{\infty}$ | $\text{EdgeBank}_{th}$ | $\text{EdgeBank}_{th}$ |
| Reddit | $\text{EdgeBank}_{\infty}$ | $\text{EdgeBank}_{th}$ | $\text{EdgeBank}_{th}$ |
| MOOC | $\text{EdgeBank}_{tw\text{-}ts}$ | $\text{EdgeBank}_{tw\text{-}re}$ | $\text{EdgeBank}_{th}$ |
| LastFM | $\text{EdgeBank}_{tw\text{-}ts}$ | $\text{EdgeBank}_{tw\text{-}re}$ | $\text{EdgeBank}_{th}$ |
| Enron | $\text{EdgeBank}_{tw\text{-}ts}$ | $\text{EdgeBank}_{tw\text{-}re}$ | $\text{EdgeBank}_{th}$ |
| Social Evo. | $\text{EdgeBank}_{th}$ | $\text{EdgeBank}_{th}$ | $\text{EdgeBank}_{th}$ |
| UCI | $\text{EdgeBank}_{\infty}$ | $\text{EdgeBank}_{tw\text{-}ts}$ | $\text{EdgeBank}_{tw\text{-}re}$ |
| Flights | $\text{EdgeBank}_{\infty}$ | $\text{EdgeBank}_{th}$ | $\text{EdgeBank}_{th}$ |
| Can. Parl. | $\text{EdgeBank}_{tw\text{-}ts}$ | $\text{EdgeBank}_{tw\text{-}ts}$ | $\text{EdgeBank}_{th}$ |
| US Legis. | $\text{EdgeBank}_{tw\text{-}ts}$ | $\text{EdgeBank}_{tw\text{-}ts}$ | $\text{EdgeBank}_{tw\text{-}ts}$ |
| UN Trade | $\text{EdgeBank}_{tw\text{-}re}$ | $\text{EdgeBank}_{tw\text{-}re}$ | $\text{EdgeBank}_{th}$ |
| UN Vote | $\text{EdgeBank}_{tw\text{-}re}$ | $\text{EdgeBank}_{tw\text{-}re}$ | $\text{EdgeBank}_{tw\text{-}re}$ |
| Contact | $\text{EdgeBank}_{tw\text{-}re}$ | $\text{EdgeBank}_{tw\text{-}re}$ | $\text{EdgeBank}_{th}$ |

## C.3 Results on Dynamic Node Classification

Table 15 shows the results of various methods for dynamic node classification on Wikipedia and Reddit (the only two datasets with dynamic labels).

## C.4 Complete Results of TCL and GraphMixer with Neighbor Co-occurrence Encoding

We show the complete results of TCL and GraphMixer with our neighbor co-occurrence encoding scheme in Table 16.

## C.5 Results of Training Time and Memory Usage Comparisons

Table 17 shows the comparisons of running time and memory usage of DyGFormer with and without the patching technique during the training process.

## C.6 Results and Discussions of Ablation Study

We validate the effectiveness of several designs in DyGFormer via an ablation study, including the usage of **N**eighbor **Co**-occurrence **E**ncoding (NCoE), the usage of **T**ime **E**ncoding (TE), and the **Mix**ing of the sequence of **S**ource node and **D**estination node (MixSD). We respectively remove these modules and denote the remaining parts as w/o NCoE, w/o TE, and w/o MixSD. We also **Sep**arately encode the **N**eighbor **O**ccurrence in the source node's or destination node's sequence and denote this variant as w/ SepNO. We report the performance of different variants on MOOC, Social Evo., UCI, and UN Trade datasets from four domains in Figure 4.

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

Our findings from Figure 4 indicate that DyGFormer achieves optimal performance when all components are utilized. The removal of any component would lead to worse results. In particular, the neighbor co-occurrence encoding scheme has the most significant impact on the performance as it effectively captures correlations between nodes. Separately encoding neighbor occurrences or encoding the temporal information could also improve performance. Mixing the sequences of the source node and destination node causes relatively minor improvements due to the usage of the neighbor co-occurrence encoding scheme because both of them aim to explore the node correlations. To this end, the necessity of designing these components has been well demonstrated.

## C.7 Results on Temporal Graph Benchmark

We also evaluate DyGFormer and baselines on the recently proposed Temporal Graph Benchmark (TGB) [23], which contains a collection of challenging and diverse benchmark datasets. We aim to verify the superiority and scalability of DyGFormer on TGB since it covers small, medium, and large datasets. We only present the analysis here. Please refer to [23] and our other work [68] for details of the datasets, tasks, evaluation metrics, experimental settings, and complete quantitative results.

**Superiority of DyGFormer**. For the dynamic link property prediction task, DyGFormer ranks first on the TGB leaderboard[7] on tgbl-wiki-v2. It ranks second/third on tgbl-coin-v2/tgbl-comment and does medium on tgbl-review-v2. We notice that the surprise index (defined as the ratio of test links that are not seen during training) on tgbl-wiki-v2 and tgbl-coin-v2 are 0.108 and 0.120, which indicates nodes tend to interact repeatedly. Such a property may be well handled by the neighbor co-occurrence

---
[7] https://tgb.complexdatalab.com/docs/leader_linkprop/

Table 14: AUC-ROC for inductive dynamic link prediction with random, historical, and inductive negative sampling strategies.

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

Table 15: AUC-ROC for dynamic node classification.

| Methods | Wikipedia | Reddit | Avg. Rank |
|---|---|---|---|
| JODIE | **88.99 ± 1.05** | 60.37 ± 2.58 | 4.50 |
| DyRep | 86.39 ± 0.98 | 63.72 ± 1.32 | 5.00 |
| TGAT | 84.09 ± 1.27 | **70.04 ± 1.09** | 4.00 |
| TGN | 86.38 ± 2.34 | 63.27 ± 0.90 | 6.00 |
| CAWN | 84.88 ± 1.33 | 66.34 ± 1.78 | 5.00 |
| TCL | 77.83 ± 2.13 | 68.87 ± 2.15 | 5.00 |
| GraphMixer | 86.80 ± 0.79 | 64.22 ± 3.32 | 4.00 |
| DyGFormer | 87.44 ± 1.08 | 68.00 ± 1.74 | **2.50** |

Table 16: AP for different methods when equipped with the neighbor co-occurrence encoding.

| Datasets | TCL | | | GraphMixer | | | DyGFormer |
|---|---|---|---|---|---|---|---|
| | Original | w/ NCoE | Improv. | Original | w/ NCoE | Improv. | |
| Wikipedia | 96.47 | **99.09** | 2.72% | 97.25 | 97.90 | 0.67% | 99.03 |
| Reddit | 97.53 | 99.04 | 1.55% | 97.31 | 97.63 | 0.33% | **99.22** |
| MOOC | 82.38 | 86.92 | 5.51% | 82.78 | 83.58 | 0.97% | **87.52** |
| LastFM | 67.27 | 84.02 | 24.90% | 75.61 | 76.48 | 1.15% | **93.00** |
| Enron | 79.70 | 90.18 | 13.15% | 82.25 | 88.83 | 8.00% | **92.47** |
| Social Evo. | 93.13 | 94.06 | 1.00% | 93.37 | 94.37 | 1.07% | **94.73** |
| UCI | 89.57 | 94.69 | 5.72% | 93.25 | 93.48 | 0.25% | **95.79** |
| Flights | 91.23 | 97.71 | 7.10% | 90.99 | 96.90 | 6.50% | **98.91** |
| Can. Parl. | 68.67 | 69.34 | 0.98% | 77.04 | 76.38 | -0.86% | **97.36** |
| US Legis. | 69.59 | 69.47 | -0.17% | 70.74 | 70.26 | -0.68% | **71.11** |
| UN Trade | 62.21 | 63.46 | 2.01% | 62.61 | 62.77 | 0.26% | **66.46** |
| UN Vote | 51.90 | 51.52 | -0.73% | 52.11 | 52.13 | 0.04% | **55.55** |
| Contact | 92.44 | 97.98 | 5.99% | 91.92 | 97.94 | 6.55% | **98.29** |
| Avg. Rank | 4.62 | 2.62 | — | 3.85 | 2.85 | — | **1.08** |

Table 17: Comparisons of training time and memory usage of DyGFormer with and without the patching technique, where OOM stands for Out-Of-Memory.

| Datasets | Input Lengths | Metrics | DyGFormer w/ patching | DyGFormer w/o patching | Reduced Ratios |
|---|---|---|---|---|---|
| LastFM | 64 | running time | 13min 58s | 21min 01s | 1.50 |
| | | memory usage | 3,945 MB | 7,953 MB | 2.02 |
| | 128 | running time | 16min 54s | 45min 29s | 2.69 |
| | | memory usage | 4,677 MB | 7,585 MB | 1.62 |
| | 256 | running time | 24min 41s | 2h 5min 50s | 5.10 |
| | | memory usage | 4,635 MB | 14,583 MB | 3.15 |
| | 512 | running time | 37min 04s | — | — |
| | | memory usage | 7,547 MB | OOM | — |
| Can. Parl. | 64 | running time | 58s | 1min 14s | 1.28 |
| | | memory usage | 2,121 MB | 3,263 MB | 1.54 |
| | 128 | running time | 1min 02s | 2min 39s | 2.56 |
| | | memory usage | 2,369 MB | 6,417 MB | 2.71 |
| | 256 | running time | 1min 26s | 6min 37s | 4.62 |
| | | memory usage | 2,855 MB | 14,923 MB | 5.23 |
| | 512 | running time | 1min 57s | — | — |
| | | memory usage | 4,511 MB | OOM | — |

encoding scheme in DyGFormer, leading to excellent performance. Correspondingly, the surprise index on tgbl-review-v2 and tgbl-comment are 0.987 and 0.823, showing that these two datasets contain many new links. This characteristic may violate the motivation of our neighbor co-occurrence encoding scheme and bring suboptimal performance. For the dynamic node property prediction task, DyGFormer also performs better than other methods with trainable parameters, but its performance is still worse than the simple non-trainable Persistent Forecast and Moving Average methods on the TGB leaderboard[8]. This observation demonstrates the necessity of proposing customized models for the dynamic node property prediction task.

**Scalability of DyGFormer**. DyGFormer can be scaled up to larger datasets that contain hundreds of thousands of nodes and tens of millions of links (e.g., tgbl-coin-v2, tgbl-comment, tgbn-genre, and tgbn-reddit) while many baselines (e.g., JODIE, DyRep, TGN, and CAWN) fail. This demonstrates the good scalability of DyGFormer.

---

[8]https://tgb.complexdatalab.com/docs/leader_nodeprop/