# OpenReview forum: "Towards Better Dynamic Graph Learning: New Architecture and Unified Library"
_NeurIPS.cc/2023/Conference — NeurIPS 2023 poster_

### Official Review · Reviewer_zX7A · 2023-07-03

**Soundness:** 3 good
**Presentation:** 4 excellent
**Contribution:** 3 good
**Rating:** 8
**Confidence:** 5

**Summary:**

This paper studies the problem of continuous-time dynamic graph learning. The authors proposed a Transformer-based architecture DyGFormer to learn the dynamic edge representation, which mainly consists of a neighbor co-occurrence encoding scheme to count the co-occurrence of nodes, a patching technique to split the node’s sequence into multiple patches, and a Transformer as the backbone.  The authors also propose a dynamic graph learning library DyGLib, and re-report the important baselines’ performance based on DyGLib. Experimental results conducted on 13 benchmark datasets verify the effectiveness of the proposed method.

**Strengths:**

1.	The authors propose a co-occurrence encoding scheme and a patching technique to capture the correlation between sequences and the long histories of sequences respectively.  As far as I know, it is less studied than learning the dynamic link representations direct from the sequences of nodes’ first-hop interactions under the dynamic graph learning problem, especially how to model the long histories of the interaction sequence.
2.	The authors conduct extensive experiments on 13 benchmark datasets to verify the effectiveness of the proposed two-component. Besides the proposed method, the authors also re-implemented the important baselines in the area, and point out that some findings of baselines are not in line with previous reports because of their varied pipelines and problematic implementations, which may benefit future research in the continuous-time dynamic graph learning community.
3.	The authors propose a continuous-time dynamic graph learning problem. This library provides a unified pipeline to train and evaluate different baselines. I think this library can provide a tool for researchers in the community to conveniently evaluate different methods in a unified setting.


**Weaknesses:**

1.	In the problem formalization of section 3, the authors define the target of the dynamic graph learning problem as learning the time-aware representations for each node. But the DyGFormer actually learns the dynamic link representations but not the node representations. Or more accurately, the DyGFormer learns the contextual node representation $h^t_{u|(u,v)}$ but not $h^t_u$, where the node representation $h^t_{u|(u,v)}$ relies on the other node v. It will be better to add some discussion for this in the paper.
2.	In section 5.2, the authors say that “some of our findings of baselines differ from previous reports”, but the authors do not point out which baselines’ performance improves. I suggest the authors explicitly list which baselines’ performance improves.


**Questions:**

Please refer to the weaknesses part.

---

> ### Author Rebuttal · Authors · 2023-08-09
>
> Thanks for your constructive reviews. In response to your comments, we have clarified the relationships between the learned representations of links and nodes. We have also specified the baselines whose observations are different from previous reports. We would be pleased to explain more if further discussions are required.
>
> **W1: It will be better to add some discussions for the learning of dynamic link representations but not node representations.**
>
> For dynamic graph learning, the status of each node is usually affected by its interactions with other nodes. Therefore, it is reasonable to learn the dynamic node representations based on their links. More concretely, for the dynamic link prediction task, future links of nodes are naturally influenced by their historical interactions. This allows us to learn from links and derive time-aware representations of source and destination nodes based on the link. For the node classification task, interactions may lead to the state change of nodes (e.g., on Wikipedia and Reddit datasets, the state of a user may change from not banned to banned because of an interaction [1]). Such a phenomenon also indicates that the dynamics of nodes can be well captured when learning from links. It is also worth noticing that many previous works (e.g., JODIE, DyRep, TGAT, TGN, …) learn in the same way as well.
>
> **W2: In section 5.2, the authors say that “some of our findings of baselines differ from previous reports”, but the authors do not point out which baselines’ performance improves. I suggest the authors explicitly list which baselines’ performance improves.**
>
> The differences between our observations and previous reports on baselines are in two main aspects.
>
> - **The performance of some baselines can be significantly improved by properly setting certain hyperparameters.** Let’s take TGAT for transductive dynamic link prediction with the random negative sampling strategy as an example (see Table 1 in our paper). Compared with the performance in [2], the results of AP are significantly improved on datasets like MOOC (from 0.61 to 0.86), LastFM (from 0.50 to 0.73), Enron (from 0.59 to 0.71), Social Evo. (from 0.76 to 0.93), Flights (from 0.89 to 0.94), and Contact (from 0.58 to 0.96). Similar improvements can also be found on JODIE, DrRep, and TGN on several datasets. This is because we perform the grid search to find the best settings of hyperparameters, which effectively improves the performance of baselines;
> - **Some methods would obtain worse results after we fix the problems or make adaptions in their implementations.** Let’s take CAWN for transductive dynamic link prediction with the random negative sampling strategy as an example (see Table 1 in our paper). Compared with the results in [2], the performance on the AP metric of CAWN drops sharply on datasets like LastFM (from 0.98 to 0.87), Can. Parl. (from 0.94 to 0.70), US Legis. (from 0.97 to 0.71), UN Trade (from 0.97 to 0.65), and UN Vote (from 0.82 to 0.53). This is because [2] uses the problematic implementations of CAWN to conduct experiments and the results will sometimes become worse after fixing the issues. You can also see Sections B.3 and B.4 in the Appendix for more detailed explanations.
>
> [1] Predicting dynamic embedding trajectory in temporal interaction networks, KDD, 2019.
>
> [2] Towards better evaluation for dynamic link prediction, NeurIPS, 2022.

---

> > ### Comment · Reviewer_zX7A · 2023-08-11
> >
> > Thanks for rebuttal, which addressed my concerns. I believe this work will advance the important dynamic graph learning field, and I am happy to increase my rating.

---

> > > ### Author Response · Authors · 2023-08-11
> > > **RE: Official Comment by Reviewer zX7A**
> > >
> > > Thank you! We appreciate your timely reply and support for our work.

---

### Official Review · Reviewer_ErNY · 2023-07-03

**Soundness:** 3 good
**Presentation:** 3 good
**Contribution:** 3 good
**Rating:** 5
**Confidence:** 4

**Summary:**

In this paper, the authors considered the dynamic graph representation learning (a.k.a. dynamic network embedding) problem and proposed a novel transformer-based architecture - DyGFormer, with several original designs (e.g., a neighbor co-occurrence encoding scheme, a patching technique, etc.) Moreover, the authors also implemented a unified continuous-time dynamic graph learning library, which include several baselines, widely-used datasets, and a common evaluation pipeline. Exhaustive experiments with various settings of dynamic link prediction and dynamic node classification were also conducted to validate the effectiveness of the proposed method.

**Strengths:**

(S1) The overall presentation of this paper is clear, which makes it easy to grasp the key ideas.

(S2) There are some original designs (e.g., neighbor co-occurrence encoding scheme and patching of historical neighbor sequence) in the proposed method.

(S3) The authors conducted many experiments on 13 datasets for both dynamic link prediction and dynamic node classification with various settings.

(S4) The authors also implemented a unified library including some typical baselines and a common evaluation pipeline for various task settings (e.g., transductive and inductive dynamic link prediction and node classification), with the code provided for review.

**Weaknesses:**

(W1) Some statements are with too many citations (e.g., 'continuous-time methods [27, 53, 62, 44, 35, 9, 55, 58, 57, 24, 34, 12]', 'graph convolutions [53, 62, 44, 35, 9, 57]', etc,) which are hard to check their sources. Some references are also repeated in successive sentences (e.g., 'only a few libraries are specifically designed for dynamic graphs [18, 45, 71]. DynamicGEM [18] is a library for dynamic graph embedding methods'). It is better to ensure that there are at most 5 references for each statement. Some of the references can also be replaced with concrete examples or methods introduced in this paper. Furthermore, references of some important statements are also missing, e.g., 'unlike most previous methods that need to retrieve nodes' historical interactions from multiple hops' but what do 'previous methods' refer to?

(W2) Some problem statements regarding model optimization are unclear. Concretely, the formal definition of training loss is not given. In general, we can divided existing dynamic graph representation techniques into the task-dependent and task-independent methods, which are respectively trained via (i) supervised losses related to and (ii) unsupervised losses regardless of the downstream task. It is unclear that the proposed method is task-dependent or task-independent. In the baselines, DyRep and TGAT are task-independent while CAW is task-dependent (with a loss designed for dynamic link prediction), as I can check. From my perspective, it is unfair to compare both types of method in a common experiment setting, where task-dependent methods are expected to have better performance than task-independent approaches, due to the incorporation of supervised information related to the downstream task. For task-independent methods, the authors should also clarify the downstream module (e.g., logistic regression, SVM, MLP, etc.) to support a concrete task. More importantly, the evaluation pipeline of a unified library should also cover both the settings.

(W3) Although the authors gave toy examples for some procedures of the proposed method (e.g., the computation of neighbor co-occurrence encoding), several details still need further clarification (e.g., how to pad a derived patch, etc.) It is recommended summarizing the overall procedure to derive all the encoding and patches in terms of pseudo-code (even in supplementary material).

(W4) In related research, inductive dynamic link prediction includes the prediction of links between (i) one old (i.e., previously observed) node and one new (i.e., previously unobserved) node as well as between (ii) two new nodes. It is unclear that the inductive setting in this paper refers to both cases or just the latter case.

(W5) Although the authors included 7 baselines in experiments, most of them are based on time-encoded deep sequential models (e.g., with RNN and attention). In addition, there are some other related approaches based on temporal point process (e.g., HTNE [1] and TREND [2]) and neural ordinary differential equation (e.g., GSNOP [3]) that are not considered in experiments. A unified library should cover various types of method.
- [1] Zuo, Yuan, et al. Embedding temporal network via neighborhood formation. KDD 2018.
- [2] Wen, Zhihao, and Yuan Fang. Trend: Temporal event and node dynamics for graph representation learning. Web Conference 2022.
- [3] Luo, Linhao, Gholamreza Haffari, and Shirui Pan. Graph sequential neural ode process for link prediction on dynamic and sparse graphs. WSDM 2023.

(W6) In supplementary material, although the authors gave some details regarding the datasets, the source information (i.e., where can we download these datasets) is missing. In Table 4 of supplementary material, what does 'duration' means? Does it means the time granularity? Consistent with the statistics of 'duration', what is the total number of time steps for each dataset? From my perspective, the scale of a dynamic graph is related to the (i) number of nodes $N$ and (ii) number of time steps $T$. As I can check, all the datasets are with $N<20,000$, which cannot be considered as large datasets. Can the proposed method be scaled up to larger datasets? Or are there any possible solutions to addressing the scalability issue (perhaps as one future research direction)?

(W7) It seems that the authors upload the paper to ArXiv before the formal submission to NeurIPS. Several month ago, google scholar and github recommended the ArXiv version of this paper to me. As a result, I have already known the names and institutions of the anonymous authors, which breaks the double-blind policy.

**Questions:**

See W1-W7.

As claimed before Eq. (3)-(7), the row dimensionality of ${\bf{Z}}^t$ (i.e., the input of transformer encoder) is $(l_u^t + l_v^t)$, which is related to the length of $S_u^t$ and $S_v^t$. Since different nodes $u$ may have $S_u^t$ with different length, it seems that one can only feed the input ${\bf{Z}}^t$ w.r.t. one link $(u, v, t)$ to the transformer encoder every time due to the non-fixed dimensionality of input. Can we simultaneously feed ${\bf{Z}}^t$ w.r.t. multiple (i.e., more than one) links $(u, v, t)$ into the transformer encoder and derive multiple embedding pairs $({\bf{h}}_u^t, {\bf{h}}_v^t)$? If yes, how to derive these embedding paris $({\bf{h}}_u^t, {\bf{h}}_v^t)$? If not, there seems to be an efficiency issue for the proposed method.

Consider an extreme case. For some new (i.e., previously unobserved) nodes without any available attributes and historical neighbors, can the proposed method still be able to support the inductive inference (e.g., dynamic link prediction and node classification) w.r.t. these nodes?

**Limitations:**

See W1-W7.

---

> ### Author Rebuttal · Authors · 2023-08-09
>
> Thanks for the detailed comments. We have clarified some presentations and provided detailed settings. We have also discussed the mentioned references and tested methods on larger datasets. We hope our answers can sufficiently address your concerns, and if that is the case, we kindly ask you to consider increasing the score.
>
> **W1: Too many or repeated citations. References of some important statements are missing.**
>
> 1) We have listed some typical methods and changed the sentences to “continuous-time methods like JODIE[27], DyRep[53], TGAT[62], TGN[44], CAWN[58], TCL[55], GraphMixer[12] and others[35, 9, 57, 24, 34]” and “graph convolutions like DyRep[53], TGAT[62], TGN[44] and others[35, 9, 57]”.
>
> 2) We clarify the former citations “…for dynamic graphs[18,45,71]” are used to list some libraries, while the latter “DynamicGEM [18]” is cited to discuss a specific one. So they don't seem to repeat.
>
> 3) We have changed the presentation to “Unlike most previous methods … hops (e.g., DyRep[53], TGAT[62], TGN[44], CAWN[58])”.
>
> **W2: Unclear training loss and downstream modules.**
>
> For fair comparisons, all models are optimized by supervised binary cross-entropy loss for dynamic link prediction and node classification tasks. In our initial attempt, we also tried to train some baselines with their own unsupervised loss(e.g., negative log-likelihood loss for temporal point process in DyRep), but they didn’t work well. For both tasks, we use MLPs as downstream modules(Lines 254-255, 261-262).
>
> **W3: Some details of the overall procedure need clarification.**
>
> Due to space limit, we elaborate on how to perform padding in a batch and we'll add an algorithm to describe the overall process in the revised paper. For a batch of multiple links, we first obtain sequences of source and destination nodes in the batch (denoted by $S_U$ and $S_V$). Then, we get the maximal sequence length in $S_U$ and $S_V$ (denoted by $l_U^{max}$ and $l_V^{max}$). If $l_U^{max}$ or $l_V^{max}$ cannot be divided by patch size $P$, we’ll increase them until they are divisible. Next, we apply zero-padding (as we use node with index 0 as padding marker) to sequences shorter than  $l_U^{max}$ and $l_V^{max}$ in $S_U$ and $S_V$. Finally, we extract encodings for the padded sequences in $S_U$ and $S_V$, divide them into  $l_U^{max}/P$ and $l_V^{max}/P$ patches, and combine them as the inputs of Transformer. Note that we'll store the original lengths of source and destination nodes, and use them to derive their final embeddings, see Equation 8.
>
> **W4: Unclear inductive setting.**
>
> A link is treated as inductive if it contains at least one new node. So the inductive setting covers both cases.
>
> **W5: Some related methods are not considered.**
>
> Thanks for providing these works! HTNE and TREND are built on Temporal Point Process(TPP), which can solve the same tasks in our work. But currently, we cannot integrate them into DyGLib since their codes are unavailable. Similar to HTNE and TREND, DyRep is also a TPP-based method that uses deep neural networks to parameterize intensity function in TPP. Our DyGFormer outperforms DyRep in the experiments, which verifies its superiority. We focus on general dynamic graphs while GSNOP is designed for sparse dynamic graphs, which can be viewed as another line of work parallel to ours. These contents will be added to the revised paper.
>
> **W6(Part 1/2): Some details of datasets are missing.**
>
> We gave the URL to download all datasets in README.md file in our codes. But we cannot show it here as NeurIPS 2023 forbids external links in rebuttal. “duration” is the time span of the dataset. For Wikipedia, Reddit, MOOC, LastFM, Enron, Social Evo. and UCI, the numbers of time steps are 152,757, 669,065, 345,600, 1,283,614, 22,632, 565,932 and 58,911. Their time granularities are Unix timestamps. For Flights, Can. Parl., US Legis., UN Trade, UN Trade and Contact, the time steps/time granularity are 122/days, 14/years, 12/congresses, 32/years, 72/years and 8,064/5 minutes.
>
> **W6(Part 2/2): Can the method be scaled up to larger datasets?**
>
> Yes. We have tested DyGFormer on two larger datasets(tgbl-review with 352,637 nodes and 4,873,540 links, tgbl-coin with 638,486 nodes and 22,809,486 links) in [1]. Due to limited time, we use hyperparameters on Wikipedia and the searches of hyperparameters are desired in future. The datasets use Mean Reciprocal Rank(MRR) for dynamic link prediction. We show results on the test set, and blank spaces mean the methods were too low or out of memory on GPU. We find DyGFormer scales well to larger datasets, while some baselines(JODIE, CAWN) fail. It does medium on tgbl-review and best on tgbl-coin, which shows its effectiveness.
>
> ||tgbl-review|tgbl-coin|
> |---|---|---|
> |JODIE|**54.97±0.91**||
> |DyRep|36.70±1.30|43.40±3.80|
> |TGAT|45.50±2.57|60.88±1.25|
> |TGN|53.20±2.00|58.30±5.00|
> |CAWN|34.43±1.39||
> |EdgeBank|9.99±0.00|58.00±0.00|
> |TCL|36.13±0.54|63.95±0.62|
> |GraphMixer|53.12±0.22|73.25±1.75|
> |DyGFormer|39.10±2.24|**74.88±0.04**|
>
> **W7: Break of double-blind policy.**
>
> Our submitted paper, appendix and codes are anonymized without identifying information. Also, NeurIPS 2023 allows authors to submit anonymized work of preprints. So we obey the double-blind policy.
>
> **Q1: Can we simultaneously feed multiple links into transformer encoder and derive multiple embedding pairs?**
>
> Yes. We use mini-batch training for DyGFormer by applying zero padding on multiple links in a batch. See **W3** for more details.
>
> **Q2: Can the method support inductive inference for nodes without attributes and historical neighbors?**
>
> DyGFormer can handle the latter case (nodes without historical neighbors) and the inductive setting already covers it (when DyGFormer predicts a new node in evaluation data). But if nodes have no attributes either, the task is hard for most existing models, which is an interesting future direction.
>
> [1]Temporal Graph Benchmark for Machine Learning on Temporal Graphs, arXiv, 2023.

---

### Official Review · Reviewer_eUNf · 2023-07-04

**Soundness:** 3 good
**Presentation:** 1 poor
**Contribution:** 2 fair
**Rating:** 4
**Confidence:** 4

**Summary:**

This paper proposes a new dynamic graph learning architecture and implemented it as part of a new unified graph library with extensive experiment results to verify the effectiveness of the proposed algorithms.

The author proposes a new Transformer-based architecture DyGFormer for dynamic graph learning. The architecture is designed to overcome some of the limitations of previous methods by:
- Leveraging a neighbor co-occurrence encoding scheme, which captures the frequency of appearances of each neighbor in the sequences of source and destination nodes. This explicitly explores the correlations between nodes.
- Utilizing a patching technique, which allows splitting each source/destination node’s sequence into multiple patches, enabling the capture of long-term temporal dependencies and reducing the computational complexity to a level that doesn't depend on the input sequence length.

The author also developed a library called DyGLib as a unified library specifically tailored to continuous-time dynamic graph learning. Its key features include:
- Standardized training pipelines to facilitate reproducibility and consistency across different methods.
- Extensible coding interfaces for better adaptability and scalability and integrates with 13 graph datasets and 9 graph algorithms

**Strengths:**

The paper presents an extensive and technically rigorous evaluation. The authors benchmark their model on 13 dynamic graph datasets against nine state-of-the-art dynamic graph learning models, providing thorough evidence for their conclusions. The comprehensive ablation study further supports the effectiveness of the proposed components.

The proposed DyGFormer architecture performs favorably in comparison to other dynamic graph learning models, with only a few exceptions.

DyGFormer handles long-term node interactions efficiently, thanks to the novel integration of neighborhood co-occurrence encoding and the patching technique.

**Weaknesses:**

The paper lacks a comprehensive technical discussion motivating the design of the proposed model. The co-occurrence matrix is said to model node correlations better than other models, but it's not entirely clear why explicit modeling of correlations leads to superior performance. Node proximity and similar neighbors are features that even basic models like the Graph Convolutional Network (GCN) can capture without explicit modeling.

The paper's presentation could be improved. The title, for instance, may overpromise what the paper actually delivers - I was expecting a unified learning algorithm along with a software tool. However, on closer reading, it appears that the main contribution is from DyGFormer, while DyGLib, though useful, doesn't seem to bring much novelty or be absolutely essential to the paper's contributions. The preliminaries section seems unfinished, and the section on DyGLib includes several unsupported claims and lacks novelty.

**Questions:**

1) Why does explicitly modeling the neighbor co-occurrence encoding help? What does the model eventually learn in the encoding module? What is the motivation of this function design which computes f(count_in_Su) + f(count_in_Sv) for each neighbor node?
2) The dynamic graph models consider the depth of graph searching but the proposed only considered first-layer interaction. Does it imply that high-level interaction modeling is not needed for dynamic graphs?

---

> ### Author Rebuttal · Authors · 2023-08-09
>
> Thanks for the helpful feedback. We have explained the motivation for our neighbor co-occurrence encoding. We have also clarified some presentations and our opinions on high-order interactions for dynamic graph learning. We are glad to answer more if there are any further issues.
>
> **W1: The paper lacks a comprehensive technical discussion …(GCN) can capture without explicit modeling.**
>
> Our neighbor co-occurrence encoding can exploit the correlations between source node $u$ and destination node $v$ by jointly learning from their sequences. However, the GCNs (involved in DyRep, TGAT and TGN) only capture relationships inside $u$/$v$’s own interactions as they compute for $u$/$v$ separately, and fail to explore the correlations between $u$ and $v$. Results of ablation study(orange bar) in Figure 4 show the need of learning correlations between $u$ and $v$. See **Q1** for an analysis of neighbor co-occurrence encoding’s motivation.
>
> **W2: The paper's presentation could be improved …unsupported claims and lacks novelty.**
>
> - **Title**. “New Architecture” means DyGFormer is a new dynamic graph learning architecture with original designs(neighbor co-occurrence encoding and patching technique), which is different from current methods. “Unified Library” means DyGLib is a unified library with standard training pipelines, extensible coding interfaces and comprehensive evaluating protocols, which is needed for dynamic graph learning.
> - **Claims of DyGLib**. DyGLib aims to solve different training pipelines(Lines 218-222), various and problematic implementations(Lines 223-229, Section B.3), and insufficient evaluation protocols(Lines 230-237) in previous studies. Empirical findings(Lines 295-300, Section B.4) well support DyGLib’s claims. Thus, DyGLib contributes to reproducible, scalable and credible dynamic graph learning research.
>
> **Q1(Part 1/2): Why does explicitly modeling neighbor co-occurrence encoding help? What does the model learn?**
>
> Our neighbor co-occurrence encoding’s motivation is more common historical neighbors imply more future interactions(Lines 170-171). To analyze its effect, we compare DyGFormer and DyGFormer without Neighbor Co-occurrence Encoding(i.e., DyGFormer w/o NCoE). We first give some symbols. $TP$, $TN$, $FN$ and $FP$ are True/False Positive/Negative. We focus on links whose predictions of DyGFormer w/o NCoE are changed by DyGFormer($FN$→$TP$, $FP$→$TN$, $TP$→$FN$ and $TN$→$FP$) and define Changed Link Ratio($CLR$) as the ratio of the changed links to their original set, which is computed by $|FN$→$TP|/|FN|$, $|FP$→$TN|/|FP|$, $|TP$→$FN|/|TP|$ and $|TN$→$FP|/|TN|$. Common Neighbor Ratio($CNR$) is the ratio of common neighbors in source node $u$’s sequence $S_u$ and destination node $v$’s sequence $S_v$, i.e., $|S_u \cap S_v|/|S_u \cup S_v|$. If our encoding is helpful, DyGFormer will fix wrong predictions of DyGFormer w/o NCoE(more $FN$→$TP$,$FP$→$TN$) and make fewer incorrect changes(fewer $TP$→$FN$,$TN$→$FP$). We compute $CLR$ and averaged $CNR$ of links in the above sets, and show results in $CLR$($CNR$) format on five typical datasets due to space limit.
>
> ||$FN$→$TP$|$FP$→$TN$|$TP$→$FN$|$TN$→$FP$|
> |---|---|---|---|---|
> |Wikipedia|68.36(18.16)|72.73(0.01)|1.68(0.10)|1.69(2.49)|
> |Flights|83.66(37.09)|83.83(2.28)|1.73(7.06)|2.11(20.28)|
> |UCI|71.45(19.08)|94.11(2.49)|7.29(3.35)|1.82(13.02)|
> |US Legis.|31.63(69.92)|23.67(62.13)|6.63(61.14)|1.59(63.80)|
> |UN Vote|44.02(78.57)|36.46(81.39)|28.95(80.86)|30.53(77.02)|
>
> We find **our neighbor co-occurrence encoding helps DyGFormer correct wrong predictions of DyGFormer w/o NCoE on datasets with several times higher $CNR$ of positive links than negative ones (this occurs on most datasets).** For Wikipedia, Flights and UCI, their $CNR$s of $FN$ are much higher than $FP$(37.09% vs. 2.28% on Flights) and DyGFormer corrects most wrong precitions of DyGFormer w/o NCoE(83.66% for positive links $FN$→$TP$, 83.83% for negative links $FP$→$TN$ on Flights). **Corrections made by our encoding are less obvious when $CNR$s between positive and negative links are similar, which only occur in 2 out of 13 datasets.** For US Legis. and UN Vote, their $CNR$s between $FN$ and $FP$ are analogous(69.92% vs. 62.13% on US Legis.), weakening our encoding’s advantage(only 31.63%/23.67% of positive/negative links are corrected in $FN$→$TP$/$FP$→$TN$ on US Legis.).
>
> Thus, we conclude neighbor co-occurrence encoding helps DyGFormer capture common historical neighbors in $S_u$ and $S_v$, and bring better results in most cases. We further compare DyGFormer with DyGFormer w/ SepNO variant that learns from $S_u$ and $S_v$ separately(orange bar in Figure 4) and get similar observations, which supports our motivation again.
>
> **Q1(Part 2/2): What’s the motivation of function f(count_in_Su) + f(count_in_Sv) for each neighbor node?**
>
> This function is designed to encode the co-occurrence features of each neighbor by combining the appearing frequencies in $S_u$ and $S_v$. Take source node $u$ as an example. Its neighbor co-occurrence features is $C_u^t\in\mathbb{R}^{|S_u^t|\times 2}$, where $C_u^t[:,0]$ is the appearing frequency of nodes $u^\prime\in S_u$ in $S_u$ and $C_u^t[:,1]$ is the frequency of nodes $u^\prime\in S_u$ in $S_v$. We first apply $f()$ on $C_*^t[:,0]$ and $C_*^t[:,1]$, and then add them to get the encoded co-occurrence features.
>
> **Q2: The dynamic graph models … Does it imply high-level interaction modeling is not needed for dynamic graphs?**
>
> Our experiments show that though DyGFormer only learns from first-hop interactions, it usually beats baselines that use high-order interactions(DyRep, TGAT, TGN…). This indicates dynamic graph learning can be well down just with first-hop interactions when we use effective designs. But we cannot say high-order interactions are not needed. Instead, it is better to infer the benefits of using high-order information in baselines are not obvious and better designs are desired to utilize such information.

---

> > ### Comment · Reviewer_eUNf · 2023-08-21
> >
> > Thanks for the detailed answers. Really appreciated your efforts on providing additional experiment results. My biggest concern of the paper which is about the weak theorical discussion and analysis is still not addressed adequately. However, given the strong empirical performance, I will revise the score from 3 to 4.

---

> > > ### Author Response · Authors · 2023-08-21
> > > **Response to Reviewer eUNf**
> > >
> > > Thanks for your valuable feedback. We understand your concern about the weak theoretical analysis of our model, which is a common challenge in the dynamic graph learning field. We have tried our best to provide an empirical discussion of our approach in the rebuttal stage, which may pave the way for future theoretical analysis. We thank you for raising your score according to our strong performance.

---

### Official Review · Reviewer_QwYr · 2023-07-06

**Soundness:** 3 good
**Presentation:** 3 good
**Contribution:** 4 excellent
**Rating:** 7
**Confidence:** 3

**Summary:**

This paper propose DyGFormer, a new Transformer-based architecture for dynamic graph learning, whose novelty mainly includes a neighbor co-occurrence encoding scheme and a patching technique. Moreover, it introduce DyGLib, a unified library to promote reproducible, scalable, and credible dynamic graph learning research. In experiments ,the DyGFormer achieves sota performance on most of the datasets.

**Strengths:**

1. The paper is clearly written and easy to follow.
2. I think this unified library is high-quality, meaningful and urgently needed, which can well promote the development of dynamic graphs


**Weaknesses:**

1. In DyGFormer , I think the introduction of transformer is not so novel. which has similarity with the attention mechanism of TGAT and TGN.
2. PINT[1], a work in NeurIPS 2022, is a new and sota model in the area of dynamic graph. Why this paper has not mention it?

[1] Souza A, Mesquita D, Kaski S, et al. Provably expressive temporal graph networks[J]. Advances in Neural Information Processing Systems, 2022, 35: 32257-32269.

**Questions:**

1. Can you add the introduction and experiment of PINT in this paper?

**Limitations:**

The authors adequately addressed the limitations. I have no ethical concerns with this paper.

---

> ### Author Rebuttal · Authors · 2023-08-09
>
> Thanks for your valuable comments. Following the reviews, we have emphasized the novelty of DyGFormer in the designs of the neighbor co-occurrence encoding scheme and patching technique. We have also added introductions and empirical comparisons with the PINT baseline. We hope our answers have sufficiently addressed your questions and we are glad to provide more answers if there are any further issues.
>
> **W1: In DyGFormer, I think the introduction of transformer is not so novel, which has similarity with the attention mechanism of TGAT and TGN.**
>
> Please kindly note that Transformer is the backbone of our DyGFormer, but it is not the essential part of our technical contributions. Instead, the novelties of DyGFormer come from the original designs of the neighbor co-occurrence encoding scheme and patching technique.
> - The neighbor co-occurrence encoding scheme explores the correlations of the source node and destination node based on their historical sequences. This point is less investigated in the existing dynamic graph learning methods because they separately compute the temporal representations of the source and destination node, and thus fail to model their correlations (e.g., JODIE, DyRep, TGAT, TGN, CAWN, …).
> - The patching technique helps DyGFormer to effectively and efficiently leverage longer histories, which is also not well addressed by previous methods.
>
> Extensive experimental results demonstrate the effectiveness of these two designs. Last but not least, we want to clarify that compared with the mentioned TGAT and TGN, our DyGFormer only needs to learn from the sequences of historical first-hop interactions of nodes, which is conceptually simple but effective. Therefore, we conclude that DyGFormer differs from existing dynamic graph learning methods with original technical contributions.
>
> **W2&Q1: PINT[1], a work in NeurIPS 2022, is a new and sota model in the area of dynamic graph learning. Why this paper has not mentioned it? Can you add the introduction and experiment of PINT in this paper?**
>
> Thanks for providing this insightful work and we apologize for not mentioning this reference. PINT [1] studies the representational power and limits of two categories of temporal graph networks, and proposes a novel architecture that is provably more expressive than both categories. Here, we provide experimental comparisons between our DyGFormer and PINT. Due to limited time during the rebuttal period, we directly compare with the official results of average precision reported in the PINT paper [1]. Please note that the comparisons are fair because the statistics and preprocessing of datasets in PINT are identical to our work.
>
> |  |  | Wikipedia | Reddit | LastFM | Enron | UCI |
> | --- | --- | --- | --- | --- | --- | --- |
> | transductive setting | PINT | 98.78 $\pm$ 0.10 | 99.03 $\pm$ 0.01 | 88.06 $\pm$ 0.70 | 88.71 $\pm$ 1.30 | **96.01 $\pm$ 0.10** |
> |  | DyGFormer  | **99.03 $\pm$ 0.02** | **99.22 $\pm$  0.01** | **93.00 $\pm$ 0.12** | **92.47 $\pm$ 0.12** | 95.79 $\pm$ 0.17 |
> | inductive setting | PINT | 98.38 $\pm$ 0.04 | 98.25 $\pm$ 0.04 | 91.76 $\pm$ 0.70 | 81.05 $\pm$ 2.40 | 93.97 $\pm$ 0.10 |
> |  | DyGFormer  | **98.59 $\pm$ 0.03** | **98.84 $\pm$ 0.02** | **94.23 $\pm$ 0.09** | **89.76 $\pm$ 0.34** | **94.54 $\pm$ 0.12** |
>
> From the results, we find that DyGFormer performs better than PINT in all the cases except for the transductive setting on UCI, which demonstrates the superiority of DyGFormer. We will include the above introductions and experiments of PINT in the revised version of our paper.
>
> [1] Provably expressive temporal graph networks, NeurIPS, 2022.

---

> > ### Comment · Reviewer_QwYr · 2023-08-17
> > **Thanks for your response**
> >
> > Thanks for your efforts to improve the paper, which have solved most of my concerns. I would like to keep the score as is.

---

> > > ### Author Response · Authors · 2023-08-17
> > > **Response to Reviewer QwYr**
> > >
> > > Thanks a lot for the response! We really appreciate your support for our work.

---

### Official Review · Reviewer_Ea5H · 2023-07-07

**Soundness:** 4 excellent
**Presentation:** 4 excellent
**Contribution:** 4 excellent
**Rating:** 7
**Confidence:** 3

**Summary:**

This paper presents a transformer-based architecture (DyGFormer) for dynamic graph learning, based on a node co-occurrence encoding scheme and patching. Further, they present DyGLib a library for uniform evaluation of dynamic graph learning techniques. Extensive experimental evaluations over diverse datasets show that DyGFormer performs well.

**Strengths:**

**Originality.** While the transformer architecture is well-known, it's application to dynamic graph learning, and the proposed co-occurrence encoding and patching schemes are novel. Various libraries / frameworks (as discussed in 108) exist for dynamic graph learning, so DyGLib is not novel in that regard, but a fresh rigorous and extensible evaluation is appreciated.

**Quality / Clarity.** The paper is well-written and easy to follow. The DyGLib codebase is high-quality and well-documented. At a glance, it looks easy to use. Also, the appendix is thorough and well-put-together.

**Significance.** Dynamic graph learning is an important research problem. This work not only presents a good solution but also paves the way for rigorous future work.

**Weaknesses:**

While DyGFormer outperforms the baselines in avg. rank, there is a problematic trend that on some datasets DyGFormer can be much worse than the best or second best baseline in terms of absolute performance points (Table 1). This needs some analysis. What characteristics of these datasets make DyGFormer a bad choice against the baselines? What aspects of DyGFormer and the baselines could be causing their poor and better performance respectively? Why does DyGFormer's superiority falter when going from "rnd" to "hist" to "ind"?

**Questions:**

Is it possible to integrate DyGLib / DyGFormer into TGL ([71] in paper)? If so, that would prevent a fragmentation of frameworks, and also allow dynamic graph learning research on billion-scale graphs.

Also, see the weaknesses section.

**Limitations:**

Limitations are discussed but in the appendix.

---

> ### Author Rebuttal · Authors · 2023-08-09
>
> Thanks for the helpful reviews. We have analyzed how the properties of datasets affect DyGFormer and explained the reasons for its varying performance under various negative sampling strategies. We have also discussed the possibility of integrating DyGLib and TGL. We hope our answers have well addressed your concerns.
>
> **W1(Part 1/2): What characteristics of datasets make DyGFormer a bad choice against baselines? What aspects of DyGFormer and baselines cause their poor and better performance?**
>
> DyGFormer is superior to baselines by 1) exploring the source and destination nodes’ correlations from their historical sequences by neighbor co-occurrence encoding, which assumes more common historical neighbors imply more future interactions(Lines 170-171); 2) using the patching technique to attend longer histories. Thus, DyGFormer tends to perform better on datasets that favor these two designs.
>
> For analysis, we give some symbols first. $TP$, $TN$, $FN$ and $FP$ are True/False Positive/Negative. Link Ratio($LR$) is the ratio of links in its corresponding set, which is computed by $TP/(TP+FN)$, $TN/(TN+FP)$, $FN/(TP+FN)$ and $FP/(TN+FP)$. Common Neighbor Ratio($CNR$) is the ratio of common neighbors in source node $u$’s sequence $S_u$ and destination node $v$’s sequence $S_v$, i.e., $|S_u \cap S_v|/|S_u \cup S_v|$. As a method with more $TP$ and $TN$(fewer $FN$ and $FP$) is better, we compute $LR$ and averaged $CNR$ of links in $TP$ and $TN$. We show results in the $LR$($CNR$) format on five typical datasets due to space limit.
>
> ||$TP$|$TN$|
> |---|---|---|
> |Wikipedia|92.74(59.09)|97.19(0.01)|
> |Flights|96.13(47.58)|95.33(1.40)|
> |UCI|82.70(28.03)|96.77(1.45)|
> |US Legis.|78.95(75.18)|56.83(53.98)|
> |UN Vote|65.18(56.24)|45.43(76.02)|
>
> We observe **when** $CNR$ **of**  $TP$ **is several times higher than** $CNR$ **of $TN$ in the datasets, DyGFormer often outperforms baselines (most datasets have this property).** For Wikipedia, Flights and UCI, their $CNR$s of $TP$ are much higher than those of $TN$(e.g., 59.09% vs. 0.01% on Wikipedia). This property matches the neighbor co-occurrence encoding’s assumption, enabling DyGFormer to correctly predict most links (92.74% of positive links and 97.19% of negative links on Wikipedia). Moreover, as LastFM and Can. Parl. can gain from longer histories (see Figure 3, Table 7), DyGFormer is significantly better than baselines on them. **When $CNR$s of $TP$ and $TN$ are less distinguishable in the datasets, DyGFormer may do worse (only 2 out of 13 datasets show this property).** For US Legis., the $CNR$ between $TP$ and $TN$ is close (75.18% vs. 53.98%), which makes DyGFormer worse than memory-based baselines (JODIE, DyRep, and TGN). For UN Vote, its $CNR$ of $TP$ is even lower than that of $TN$(56.24% vs. 76.02%), which is opposite to our assumption, making DyGFormer perform worse than a few baselines. Since these two datasets don’t obviously gain from longer sequences either (see Table 7), DyGFormer cannot achieve satisfactory results on them.
>
> **W1(Part 2/2): Why does DyGFormer's superiority falter when going from "rnd" to "hist" to "ind"?**
>
> Compared with rnd, hist and ind will sample previous links as negative ones. This makes previous positive links negative, which tends to hurt DyGFormer since the assumption of neighbor co-occurrence encoding scheme may be violated. As positive links are the same among rnd, hist, and ind, we compute $LR$ and averaged $CNR$ of links in $FP$ and show results in the $LR$($CNR$) format.
>
> |||$FP$|
> |---|---|---|
> |Wikipedia|rnd|2.81(0.02)|
> ||hist|89.28(14.00)|
> ||ind|94.53(11.66)|
> |Flights|rnd|4.67(0.01)|
> ||hist|94.52(35.62)|
> ||ind|92.94(30.29)|
> |UCI|rnd|3.23(9.98)|
> ||hist|64.93(12.22)|
> ||ind|76.42(13.81)|
> |US Legis.|rnd|43.17(79.40)|
> ||hist|17.31(87.51)|
> ||ind|21.21(75.51)|
> |UN Vote|rnd|54.57(79.60)|
> ||hist|39.90(75.53)|
> ||ind|52.92(79.15)|
>
> We find **when hist or ind causes several times higher** $CNR$ **of** $FP$ **than rnd in the datasets, DyGFormer drops sharply.** For Wikipedia, Flights and UCI, the $CNR$s of $FP$ with hist/ind are much higher than rnd (14.00%/11.66% vs. 0.02% on Wikipedia). This misleads DyGFormer to predict negative links as positive and causes drops (89.28%/94.53% of negative links are incorrectly predicted with hist/ind on Wikipedia, while only 2.81% are wrong with rnd). We also note the drops on UCI are milder since the changes in $CNR$ caused by hist or ind vs. rnd are less obvious than changes in Wikipedia or Flights. **When changes in** $CNR$ **of** $FP$ **caused by hist or ind are not obvious in the datasets, DyGFormer is less affected.** As hist/ind makes little changes in $CNR$s of $FP$ on US Legis., we find it ranks second with hist/ind, which may indicate DyGFormer is less influenced by neighbor co-occurrence encoding on US Legis. and generalizes well to various negative sampling strategies. For UN Vote, though its $CNR$s of $FP$ are not affected by hist and ind either, DyGFormer still performs badly due to its inferior performance with rnd.
>
> To this end, we conclude that for datasets with much higher $CNR$ of positive links than negative ones or datasets that can benefit from longer histories, DyGFormer is a good choice. Otherwise, we may need to try other methods. Also, our neighbor co-occurrence encoding may be sometimes fragile to various negative sampling strategies if its assumption is violated. It’s promising to tackle this as future work.
>
> **Q1: Is it possible to integrate DyGLib/DyGFormer into TGL?**
>
> Yes. Technically, TGL trains on large-scale dynamic graphs with some engineering implementations (e.g., realizing temporal neighbor sampler by C++ for efficient sampling, using multiple GPUs for parallel training). Our DyGLib contains more popular dynamic graph learning models and supports comprehensive evaluations. Hence, DyGLib and TGL are complementary and basically compatible as most of their modules are developed by PyTorch. It is promising to combine them for a better framework.

---

> > ### Comment · Reviewer_Ea5H · 2023-08-19
> >
> > Thanks for the rebuttal. I would like to keep my rating.

---

> > > ### Author Response · Authors · 2023-08-20
> > > **Response to Reviewer Ea5H**
> > >
> > > Thanks for your response! Your support for our work means a lot to us.

---

### Author Rebuttal · Authors · 2023-08-09

We would like to thank all the reviewers for their valuable feedback and helpful comments on our work. We are delighted by the reviewers’ acknowledgments that the proposed DyGFormer is novel with original designs (Reviewers Ea5H, eUNf, ErNY, and zX7A), the presented DyGLib is of high quality and advances the dynamic graph learning field (Reviewers Ea5H, QwYr, and zX7A), the experimental evaluations are rigorous and extensive (Reviewers Ea5H, eUNf, ErNY, and zX7A), and the paper is well-organized and easy to follow (Reviewers Ea5H, QwYr, ErNY, and zX7A).

To the best of our efforts, we have provided thorough responses to address the concerns raised by each reviewer. In particular, the responses mainly consist of:

- Analysis of why the performance of DyGFormer varies across different datasets and various negative sampling strategies;
- Additional introductions and empirical comparisons with the PINT baseline;
- Explanations of the motivation of the neighbor co-occurrence encoding scheme;
- Discussions of several related dynamic graph learning methods based on temporal point process and neural ordinary differential equation;
- Additional evaluations of DyGFormer and baselines on larger datasets;
- Specifications of some details in experimental settings;
- Elaborations of certain experimental analysis.

In summary, all the comments are very valuable for us to improve the quality of this work and we will incorporate them in the revised version of our paper. Once again, we thank the reviewers for their constructive feedback.

---

### Decision · Program_Chairs · 2023-09-21

**Decision:**

Accept (poster)

**Comment:**

This paper introduces an innovative approach to dynamic graph learning through the DyGFormer architecture and DyGLib library. Reviewers appreciate the originality, clear presentation, and significance of the work in addressing an important research problem. DyGFormer's strengths lie in its novel features, such as co-occurrence encoding and patching, which contribute to its efficient handling of long-term node interactions.

Reviewers raise concerns about technical discussions motivating the proposed model's design, as well as the need for clearer explanations and addressing technical issues. However, the value of the contributions, coupled with the extensive evaluation, outweighs these concerns. Acceptance is recommended, and it is encouraged to integrate the reviewers' feedback during the preparation of the final version of the paper.